# A DDoS Attack Detection Method Based on Natural Selection of Features and Models

**Ruikui Ma** [1,2,3]**, Xuebin Chen** [1,2,3,]*[iD] **and Ran Zhai** [1,2,3]

1 College of Science, North China University of Science and Technology, Tangshan 063210, China
2 Hebei Key Laboratory of Data Science and Application, Tangshan 063210, China
3 Tangshan Key Laboratory of Data Science, Tangshan 063210, China
* Correspondence: chxb@ncst.edu.cn

**Abstract:** Distributed Denial of Service (DDoS) is still one of the main threats to network security today. Attackers are able to run DDoS in simple steps and with high efficiency to slow down or block users' access to services. In this paper, we propose a framework based on feature and model selection (FAMS), which is used for detecting DDoS attacks with the aim of identifying the features and models with a high generalization capability, high prediction accuracy, and short prediction time. The FAMS framework is divided into four main phases. The first phase is data pre-processing, including operations such as feature coding, outlier processing, duplicate elimination, data balancing, and normalization. In the second stage, 79 features are extracted from the dataset and selected by the feature selection algorithms filter, wrapper, embedded, variance, mutual information, backward elimination, Lasso.L1, and random forest. The purpose of feature selection is to simplify the model, avoid dimensional disasters, reduce computational costs, and reduce the prediction time. The third stage is model selection, which aims to select the most ideal algorithm from GD, SVM, LR, RF, HVG, SVG, HVR, and SVR using a model selection algorithm for the selected 21 features, and the results show that RF is far ahead in all evaluation indexes compared to the other models. The fourth stage is model optimization, which aims to further improve the performance of the RF algorithm in detecting DDoS attacks by optimizing the parameters max_samples, max_depth, n_estimators for the initially selected RF by the RF optimization algorithm. Finally, by testing the 100,000 CIC-IDS2018, CIC-IDS2017, and CIC-DoS2016 synthetic datasets, the results show that all the results have achieved excellent performance in the same category. Moreover, the framework also shows an excellent generalization performance by testing over 1 million synthetic datasets and over 330,000 CIC-DDoS2019 datasets.

**Keywords:** DDoS attacks; machine learning; feature selection; ensemble learning; random forest





## 1. Introduction

Distributed Denial of Service (DDoS) attacks typically combine multiple computers to launch an attack against a target in order to increase the power of the attack, deny legitimate access to normal users by exhausting bandwidth and resources, and also affect the overall performance of the network [1]. DDoS attacks have been used as an information warfare tool in warfare [2]. Since DDoS attacks use the same legitimate network protocols as normal users, it is difficult to distinguish DDoS attackers from normal users from huge network traffic by a single protocol or feature alone. Coupled with unreasonable feature selection methods and detection models, DDoS attacks are even less easy to detect [3]. There are many tools that can generate DDoS attacks quickly and easily, such as trinoo, Low Orbit Ion Cannon, Trinity, mstream, Tribe Flood Network, etc. These tools differ in terms of architecture, types of flooding attacks, and the methods used for DDoS attacks [4].

With the rapid development of machine learning (ML) in computer vision (CV), natural language processing (NLP), robotics, image processing, and many other fields [5],

the application of machine learning techniques in cyber security helps machines to make correct decision analyses and predictions [6]. Machine learning-based techniques are effective in the application of detecting and distinguishing DDoS attacks. Some of the available DDoS attack detection techniques include artificial neural nets (ANNs), Bayesian network (BN), gradient descent (GD), decision tree (DT), ensemble learning (EL), random Ffrest (RF), logistic regression (LR), and support vector machine (SVM) [7]. In this paper, we propose FAMS, a DDoS attack detection framework based on machine learning, which is divided into four phases: the data preparation phase, the feature selection (FS) phase, the model selection (MS) phase, and the RF optimization phase. Many works within the literature do not focus on pre-processing the data [8], which may eventually lead to problems such as poor model accuracy and a weak generalization ability. Pre-processing the data in advance can avoid noise, enhance generalization, improve algorithm accuracy, and reduce the prediction time. The data preparation stage includes operations such as feature extraction, feature coding, missing value filling, outlier removal, data balancing, duplicate elimination, and normalization. It is not always the case that the more data there is, the better the results. Irrelevant features only overwhelm the learning process and tend to over-fit the model, increasing the prediction time. Generally, FS methods can be divided into three categories, namely, (1) filter [9], (2) wrapper [10], and (3) embedded [11], with numerous FS methods in each category. We include all three types of methods in the model selection phase of our proposed FAMS framework and, in addition, we propose a feature selection algorithm that uses numerous FS techniques to extract the best combination of features in the dataset. By combining them in a way that removes the inherent biases and drawbacks of using them individually, the DDoS detection model is better simplified, irrelevant features are removed, data dimensionality is reduced, dimensional catastrophes are avoided, computational costs are reduced, model prediction time is reduced, and model generalization is enhanced to avoid causing overfitting. There are many machine learning methods, and machine learning can usually be classified into four categories: supervised learning, unsupervised learning, semi-supervised learning, and reinforcement learning [12]. Each category, in turn, contains numerous algorithms. No single machine learning algorithm can achieve the best results for any dataset or any data feature. Before choosing a machine learning model algorithm, it is often necessary to consider the size of the dataset, the features, and the nature of the problem to be solved. If an inappropriate machine learning model algorithm is chosen, not only will it fail to yield accurate results, but it will lead to overfitting of the model [13], requiring longer training and prediction times, and thus, failing to provide effective DDoS attack detection for realistic networks with high rates of high traffic. A reasonable choice of machine learning model algorithms can improve accuracy, reduce prediction time, and enhance the generalization capability of the model. This is especially true in realistic network systems with high density and high speed rates. Anomalous traffic can be detected quickly and effectively through the judicious use of machine learning algorithm models. Therefore, we selected the most suitable algorithm from GD, SVM, LR, RF, HVG (GDs hard voting algorithm), SVG (GDs soft voting algorithm), HVR (RFs hard voting algorithm), and SVR (RFs soft voting algorithm) according to the model selection algorithm in the model selection phase. The model selection algorithm found that RF outperformed the other models in terms of Accuracy, Precision, Recall, F1_Score, Average, and predict_time. Therefore, in the subsequent RF optimization phase, we choose RF as the object of optimization improvement, and the model will be further improved in performance by the RF optimization algorithm.

In this study, we propose the FAMS framework, which first pre-processes the CIC-DDoS2019, CIC-IDS2018, CIC-IDS2017, and CIC-DoS2016 datasets in the data preparation phase. In the feature selection stage, the best 21 feature combinations were selected by the feature selection algorithm. In the model selection stage, the optimal algorithm RF was initially selected by the model selection algorithm. Then, the algorithm was optimized to further improve the DDoS attack detection performance of the algorithm through the RF optimization stage. Finally, by testing on 100,000 CIC-IDS2018, CIC-IDS2017, and CIC-

DoS2016 synthetic datasets, on 1,000,000 synthetic datasets, and 330,000 CIC-DDoS2019 datasets, the experimental results show that our framework FAMS compares with DDoS detection model frameworks in the same category in terms of Accuracy, Precision, Recall, F1_Score, Average, and predict_time. The results also show that the framework also exhibits an excellent generalization performance when tested on the 1 million synthetic dataset and the 330,000 CIC-DDoS2019 dataset.

The rest of the paper is organized as follows: In Section 2 we review related work by others on DDoS attack detection. Section Section 3 describes the data pre-processing, FS algorithm, MS algorithm, and finally, the RF optimization algorithm in the FAMS framework. In Section Section 4, the experimental results are analyzed. Section Section 5 concludes the paper and provides an outlook for the future.

## 2. Related Works

With the continuous development of artificial intelligence, machine learning plays an important role in DDoS attack detection. Nanda et al. [14] proposed four machine learning algorithms based on C4.5, plain Bayesian, decision trees, and Bayesian networks to train historical network attack data to detect network attacks and finally found that an average prediction accuracy of 91% was achieved by using Bayesian networks. Fukuda et al. [15] proposed machine learning algorithms based on regression trees, classification, support vector machines, and random forests to classify the activities of malicious traffic originators by using the Domain Name System (DNS) backscatter as an additional source of information about network activities, and finally achieved an accuracy of about 75%. Deepa V. et al. [16] proposed machine learning algorithms based on plain Bayes, support vector machines, K-nearest neighbors, and self-organizing map (SOM) integration techniques for DDoS attack detection. Finally, the integrated approach was found to exhibit the best results. Sharafaldin et al. proposed a new detection and family classification method based on network flow features by detecting the CIC-DDoS2019 dataset and finally providing the weights of each feature of the dataset. Zhong et al. [17] proposed a big data-based hierarchical deep learning system in order to improve the intrusion detection rate (BDHDLS), which reduces the construction time of the model BDHDL by analyzing the features and behavior of the data and using distributed parallel training techniques. Qu et al. [18] proposed an improved neural network model that identifies abnormal users by analyzing web logs, which was eventually found through experiments to have a better performance than the traditional SVM and LSTM models. Cil et al. [19] proposed a deep neural network (DNN)-based machine learning algorithm for packet capture detection from DDoS attacks in network traffic and achieved a 94% attack classification accuracy by detecting the CIC-DDoS2019 dataset. Khempetch et al. [20] proposed a deep neural network (DNN) and long short-term memory (LSTM) algorithms and introduced a new DDoS attack classification method, which was found to achieve good detection results by detecting the CIC-DDoS2019 dataset. Hosseini and Azizi [21] proposed a hybrid model based on data flow to detect DDoS attacks. The task was organized by separating the computation of resources on the agent side and the client side. Decision trees, random forests, Multi-Layer Perception (MLP), k-nearest neighbors, and plain Bayes were used to identify DDoS attacks and finally, random forests were found to have the best results. Yong et al. [22] used a principal component analysis (PCA) for feature selection. Dash et al. [23] summarized and classified a number of feature selection methods through a survey and selected a representative feature selection method from each of the special selection methods, and then explained the advantages and disadvantages of the different feature selection methods. Chandrashekar et al. [24] provided a detailed survey of various FS methods, focusing on filters, wrappers, and embedded feature selection methods. The authors demonstrated the usefulness of special selection through experiments on standard datasets. Sheikhpour et al. [25] presented a survey of semi-supervised feature selection algorithms by presenting two semi-supervised selection methods for classification. Khalid et al. [26] investigated a large number of FS methods and then experimentally checked the applicability of different FS and feature extraction

techniques, analyzing how these methods are effective in improving the performance of the algorithms. Martinez et al. [27] (2021) conducted an experiment with feature selection and different decision tree-based algorithms to select RF and XGBoost in the F-measure for a complete feature set detection, and achieved 98.5% high performance and ultimately found that choosing the right features and machine learning algorithms, i.e., using fewer attributes for network intrusion detection systems (NIDS), did not significantly degrade the performance.

This section provides a review of the literature related to DDoS intrusion detection. In summary, it is found that in recent years, although deep learning, a subclass of machine learning, has gained increasing popularity among researchers and is widely used for network intrusion detection [28], deep learning requires long training and detection times due to its complexity and the need for large amounts of data for training. Although most researchers use various FS methods to process multiple types of datasets, they rarely consider integrating feature selection using more than two types of feature selection methods [29]. In addition, there are different feature selection methods and numerous machine learning methods available for the detection of DDoS attacks. Unselected, optimized, and single feature selection methods or machine learning algorithms may not achieve satisfactory results. After data pre-processing, feature selection plays an important role in the overall DDoS detection system as the basis of machine learning. A wrong feature selection method not only fails to improve the overall performance, but also falls into the curse of dimensionality by introducing too many redundant features [30]. This makes it difficult to improve the final detection rate of DDoS attacks no matter what machine learning methods are subsequently used to optimize the detection of DDoS. Detection is based on feature selection, and because of the many machine learning methods, blindly selecting one or more machine learning methods may also lead to the same problems as incorrect feature selection, such as low accuracy of DDoS attack detection, long detection time, and low generalization performance.

In this paper, we propose a dual feature and model selection-based DDoS attack detection framework, FAMS, which draws on the "survival of the fittest" rule in biology. By processing the latest DDoS datasets from a large number of different sources and using a grid algorithm to select various features and machine learning algorithms, we find that the FAMS framework selects the optimal combination of 21 features through a feature selection algorithm. The FAMS framework was experimentally found to have selected the optimal combination of 21 features using the feature selection algorithm. Then, the most desirable algorithm, RF, was initially selected from GD, SVM, LR, RF, HVG, SVG, HVR, and SVR by the model selection algorithm. Finally, the performance of DDoS attack detection was further improved by the RF optimization algorithm for the initially selected RF. The results were excellent in terms of detection time and generalization performance.

## 3. Materials and Methods

This section provides an overview of our proposed integration framework FAMS. Figure 1 gives a detailed architecture diagram describing the process flow. It consists of four parts, the data preparation phase, the feature selection phase, the model selection phase, and the model evaluation phase.

### 3.1. Dataset

The dataset is the most fundamental part of FAMS, and the novelty of the selection of the dataset is directly related to the subsequent feature and model processing of the framework. There are two main types of DDoS datasets in this paper. One is from CIC-DDoS2019 [31]. The other is from DDoS streams and corresponding 'BENIGN' streams that are extracted from three different datasets from CIC-IDS2018 [32], CIC-IDS2017 [33], and CIC-DoS2016 [34] at different times and with different DDoS attack methods, and are then synthesized into a synthetic dataset with a total of 84 dimensional features and 10 million records. As a result, this synthetic dataset has more records and more diverse features than

a single dataset, and better reflects the comprehensive performance of the model in terms of noise immunity, generalization capability, and accuracy. Each dataset is briefly described below.

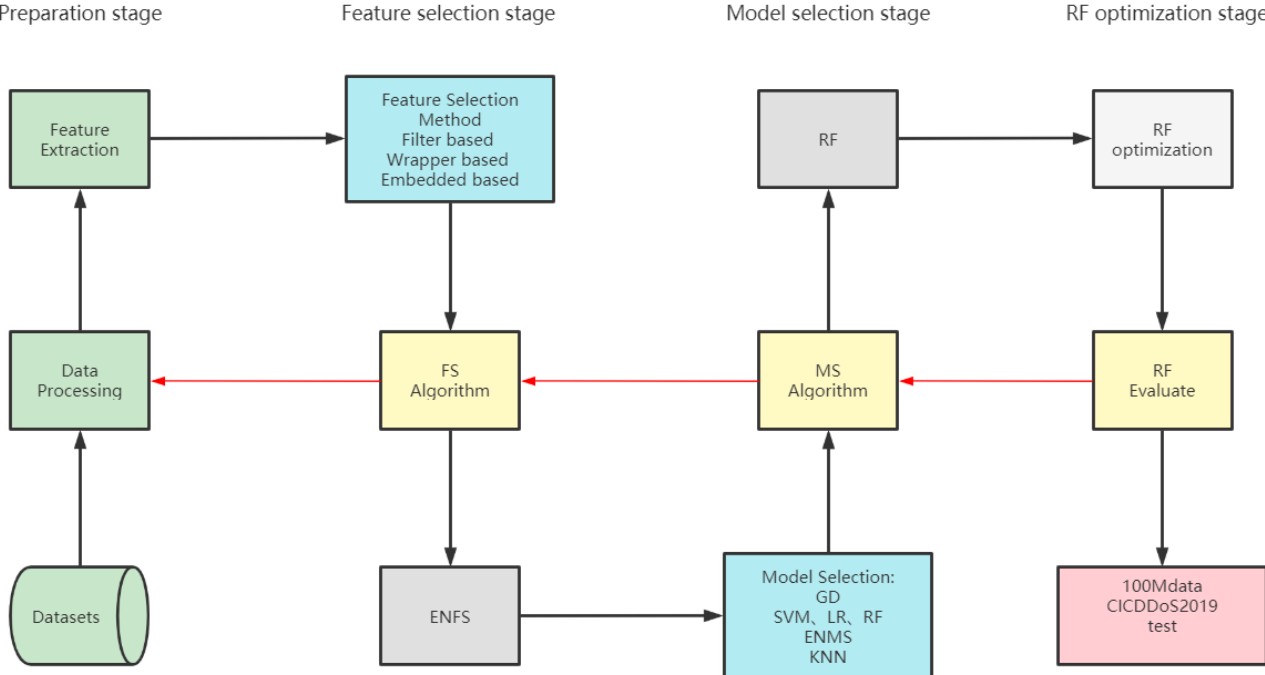

**Figure 1.** Framework for feature selection and model selection methods (FAMS).

The CIC-IDS 2018 dataset contains 80-dimensional network flow features containing seven different attack scenarios of Heartbleed, Web Attack, DoS, DDoS, Internal Penetration [35], Botnet [36], and Brute Force. The CIC-IDS 2017 dataset contains 80+ dimensional network flow features containing attack types of Botnet, DoS, DDoS, and Web. The CIC-DoS2016 dataset contains 84-dimensional network flow characteristics, with four attack types and eight application-layer dos attack traces, with attack types skewed toward slow send and slow read slow attacks. The CIC-DDoS2019 dataset is the most comprehensive and up-to-date DDoS attack dataset that is publicly available to date, providing 87-dimensional network flow characteristics including protocol, destination port, timestamp, etc., and contains LDAP, PORTMAP, NTP, MSSQL, SSDP, CharGen, NetBIOS, UDP Flood, UDP Lag, DNS, TFTP, SNMP, and Syn Flood [37], a total of 13 different types of DDoS attacks.

In the data pre-processing stage, it includes operations such as feature coding, outlier processing, duplicate elimination, data balancing, and normalization. Some algorithms do not work when there are missing values, and since our dataset of over 10 million is very large and the number of missing values is small, the missing value records are removed [38]. Although the presence of outliers tends to introduce noise into the dataset, reducing the representativeness of the sample and leading to overfitting of the model, our model is an RF consisting of numerous decision trees, which are more robust to datasets containing outliers and have higher generalization performance [39]. We will also normalize the data in order to eliminate the different magnitudes of the different traffic features and to reduce the computational overhead. Finally, we divide the dataset into a training set for model training and a test set for model prediction according to 70% and 30%.

### 3.2. Feature Selection Methods

Feature selection is the process of selecting the most relevant and beneficial features to improve the prediction of the model itself during model construction, and FS is a very important part as the foundation in this framework FAMS. Too many redundant features will only increase the computational overhead of the model, increase the risk of model

overfitting, and greatly increase the training and detection time of the model. With feature selection, dimensional catastrophes can be avoided, making the model simpler and easier to understand, with fewer computational costs and shorter training and prediction times, thus enhancing model generalization. Feature selection and model training are not two separate processes, and different combinations of features can provide different performance for different DDoS detection algorithms. Each of the three main categories of FS methods is briefly discussed in the next section, and the feature selection algorithm section of this FAMS framework is presented at the end of this section.

### 3.2.1. Filter

The filter method does not care about the ML algorithm used. Subsequently, it is based on performance measures to select features. The common feature selections are variance, mutual information, information value, Chi-square, and correlation. Filter calculates the correlation value between each feature and the target variable, and selects features by judging whether the correlation value exceeds a threshold [40]. The main feature of filter is the low computational overhead, but it leads to lower prediction accuracy of the model.

(a)  Variance

Variance is used to filter features by their variance. When a feature has a small variance on its own, it means that the feature has very little variance, and then the feature is of little use in differentiating the dataset. Therefore, before feature engineering begins, features with a variance of zero are generally eliminated to reduce the number of subsequent calculations.

(b)  Mutual Information

In probability theory, the mutual information of two random variables is a measure of the interdependence between the variables, and the mutual information of two discrete random variables X and Y can be defined as in Equation (1). The mutual information is not restricted to real-valued random variables; it is more general and determines how similar the product p(X)p(Y) of the joint distribution p (X, Y) and the decomposed marginal distributions are. From Equation (1), the larger the value of the mutual information, the greater the mutual dependence between the features, which means that they are "more relevant", and vice versa. Mutual information is one of the filtered feature selection algorithms that can be used to remove redundant features in feature selection, thereby obtaining a subset of features that better describe the given problem with minimal performance loss [41].

$$I(X;Y) = \sum_{x,y} p(x,y) \log \frac{p(x,y)}{p(x)p(y)} \tag{1}$$

where p(x) is the probability of $X = x_i$ and p(y) is the probability of $Y = y_i$, p(x.y) is the joint probability, i.e., the probability of $X = x_i$ and $Y = y_i$ occurring simultaneously, where the base of the log can be chosen as e or 2.

### 3.2.2. Wrapper

Wrappers look for all feature subsets and evaluate the selection of a high performance subset using the evaluation metrics of the ML algorithm [42], commonly backward elimination, grid search, and forward selection. Since wrapper retrains a new model on each feature subset, the model computation overhead is relatively high and also requires the definition of a condition for feature selection to stop, which can be when the number of features one needs has been reached or when performance starts to degrade, etc.

Backward Elimination

The main principle of backward elimination is that it will first start with all features and will perform ML algorithm evaluation, then, after removing a feature, it will perform ML algorithm evaluation again, and so on. Backward elimination will try to remove each

feature and perform evaluations to test which feature has the biggest improvement on the model accuracy until the required stopping condition is reached [43].

### 3.2.3. Embedded

Embedded combines filter and wrapper. Embedded is feature selection and algorithm training in parallel, and is a method of letting the algorithm itself decide which features to use [44]. Common embedded include Lasso and a range of tree-based algorithms. It is characterized by the fact that embedded is less computationally expensive than wrapper because it only trains the model once and takes into account the interaction between features.

### Lasso

Lasso avoids model overfitting by adding penalties to the parameters of the ML model. The basic idea is to control the degrees of freedom of the model by adjusting the parameters t as shown in Equations (2) and (3) below.

$$B_{LASSO} = \arg_B \min\{|Y - \sum_{j=1}^{p} X_j B_j|\} \tag{2}$$

$$\text{s.t.} \sum_{j=1}^{p} |B_j| \leq t \tag{3}$$

arg is the parameter, B is the required parameter, j is the sample index, X is the input data, s.t. is the constraint, P is the total number of samples, and t is the penalty term to control the complexity of the model and avoid over-complexity of the model.

Typically L1, L2, and L1/L2 are used for a total of three regularizations for linear models, while Lasso (L1) has the property of being able to reduce some coefficients to zero. lasso regression has a cost function as in Equation (4). The essence is to adjust $\lambda$ to achieve a balanced adjustment of the model error and variance [45].

Cost function for LASSO regression:

$$J(\theta) = \frac{1}{2} \sum_{i}^{m} \left(y^{(i)} - \theta^T x^{(i)}\right)^2 + \lambda \sum_{j}^{n} |\theta_j| \tag{4}$$

m is the number of samples, n is the number of features, $\theta$ is the parameter vector, $\theta^T$ is the transpose of $\theta$. $x^{(i)}, y^{(i)}$ is the coordinate vector of the ith instance, and $\lambda$ is the adjustment parameter by which $\lambda$ is adjusted to achieve a balanced adjustment of the model error and variance.

Matrix form:

$$J(\theta) = \frac{1}{2n}(X\theta - Y)(X\theta - Y) + \alpha ||\theta||_1 \tag{5}$$

X is a m × n matrix, Y is m × 1 vector, $\theta$ is a n × 1 vector, n is the number of samples, $\alpha$ is a constant and needs to be tuned, and $||\theta||_1$ is the norm.

Therefore, for linear and logistic regression, when using Lasso regularization to remove minor features, it is important to reasonably increase the penalty to avoid removing important features from the data by increasing the penalty excessively, but insufficient penalties can also lead to too much redundant data [46].

In summary, the feature selection algorithm is the basis of the FAMS algorithm and plays a crucial role in the subsequent model selection. The aim is to use a feature selection method from a large number of features in the dataset, from which a feature selection method with high generalization capability, high performance metrics, and short prediction time is selected. Specifically, as follows, first we apply the grid algorithm [47]: for Filter-Variance, Filter-Correlation, Filter-Chi-Square, Filter-MutualInformation, Filter-Information, and Embedded from Filter, Wrapper, and Filter-InformationValue, Wrapper-

ForwardSelection, Wrapper-BackwardElimination, Wrapper-Exhaustive, Wrapper-Genetic, Embedded-Lasso, Embedded Embedded-RandomForest, Embedded-GradientBoosted, and Embedded-Recursive for a total of 13 feature selection methods, using 10MCIC-IDS2018, CIC-IDS2017, and CIC-DoS2016 synthetic DDoS datasets, choosing the common model algorithms GD, SVM, LR, and RF, and then, model evaluation metrics Accuracy, Precision, Recall, and F1_Score for performance evaluation, and take the average value, Average, which is used for ranking as a reference for selecting models as in the following Equation (6).

$$\text{Average} = \frac{Accuracy + Precision + Recall + \text{F1\_Score}}{4} \tag{6}$$

The five most suitable feature selection models for DDoS detection using the Algorithm 1: feature selection algorithm are variance, mutual information from filter, backward elimination from wrapper and Lasso.L1, and random forest from embedded. These five feature selection methods are from each of the three types (filter, wrapper, and embedded) and are included in the feature selection algorithm of our proposed FAMS framework as a way to eliminate the inherent biases and drawbacks associated with one of them when used alone. This is conducted in a way that eliminates the inherent biases and drawbacks associated with one of the classes when used alone, to differentiate classes, reduce feature bias, and enhance the generalization performance of the features selected by the feature selection algorithm to avoid overfitting. The algorithm pseudo-code is shown in Algorithm 1: feature selection algorithm. The flow chart of the algorithm is shown in Figure 2.

---

**Algorithm 1**: Feature Selection Algorithm

---

**Input:** dataset = [CIC-IDS2018, CIC-IDS2017, CIC-DoS2016],
                features = [Filter-Variance, Filter-Correlation, Filter-Chi-Square Filter-MutualInformation,
Filter-InformationValue, Filter-InformationValue Wrapper-ForwardSelection, Wrapper-BackwardElimination, Wrapper-Chi-Square
                Wrapper-Exhaustive, Wrapper-Genetic, Embedded-Lasso.
                Embedded-RandomForest, Embedded-GradientBoosted, and Embedded-Recursive]
                model = [GD, SVM, LR, RF]
**Output:** Select 21 most important features.
**procedure:** Feature Selection
      Step 1: defined results = [Method, matrix, Accuracy, Precision,
                       Recall, F1_Score, Average, predict_time]
      Step 2: defined method features_method_results (results) # select 5 FS algorithms
      Step 3: for each ds in dataset:
           for each fts in features:
                    model.fit
               testes_out(y_test,y_pred) # Performance test function
                   results.append(tests_out(y_test,y_pred))
      Step 4: features_method_results(results)# Apply the method to get 5 feature selection algorithms
      # The following is the feature selection phase using 5 feature selection algorithms
      Step 5: defined variable selected_features_ds_fts = [ ]
      Step 6: dataset = [CIC-IDS2018, CIC-IDS2017, CIC-DoS2016]
      Step 7: features = [Variance, Mutual Information, Backward Elimination,
                 Lasso.L1, Random Forest]
      Step 8: for each ds in dataset:
        for each fts in features:
                  fts.selection_feature #Count the features selected by each method
          selected-features-ds-fts.append(fts.selected_feature)
      Step 9: defined variable feature_selection_results = [ ]
      Step 10: for each feature in selected_features_ds_fts:
                If(feature > 3):
                    feature_selection_results.append(feature)
      Step 11: feature_selection_results
**end procedure**

---

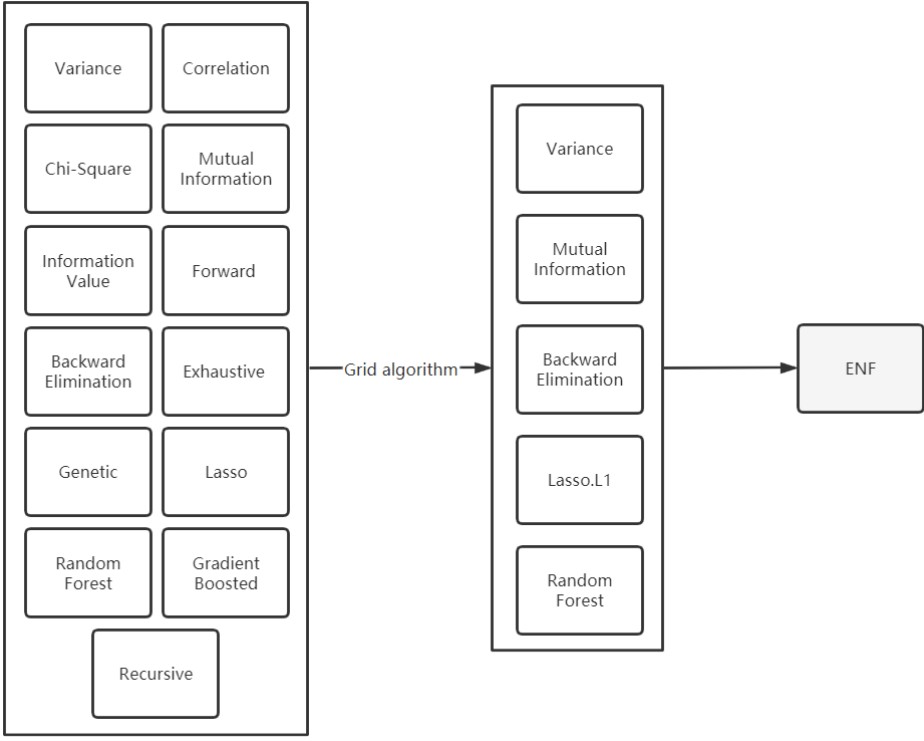

**Figure 2.** Basic principle of feature selection algorithm.

*3.3. Model Selection Methods*

Machine learning is used to classify things, discover patterns, predict outcomes, and make informed decisions. The many machine learning methods can be broadly classified into four types, namely, supervised learning, semi-supervised learning, unsupervised learning, and reinforcement learning [48]. Each of these models in turn contains multiple algorithms. No single machine learning model algorithm can achieve the best results for any dataset and any data feature. Usually, the size of the dataset, the characteristics of the dataset, and the nature of the problem to be solved need to be taken into account before a machine learning model algorithm is selected [49]. If an inappropriate machine learning model algorithm is selected, not only will it not yield accurate results, but it will lead to overfitting of the model or will require longer training and prediction time for effective DDoS attack detection on realistic high speed and high traffic networks. A reasonable choice of machine learning model algorithms can improve accuracy, reduce prediction time, and enhance the generalization ability of the model. Especially in realistic network systems with high-density data flow, anomalous traffic can be detected quickly and effectively through the judicious use of machine learning algorithm models [50]. In this section, the characteristics of the machine learning methods used in this study are first reviewed. Then, the section will conclude with a description of the model selection algorithm component of the FAMS framework.

3.3.1. GD

The basic principle of gradient descent is to iteratively adjust the parameters in order to minimize the cost function. That is, you start with a random initialization using a random value of θ and then gradually improve, taking one step at a time, with each step trying to reduce the cost function a little (e.g., MSE) until the algorithm converges to a minimum value. To implement gradient descent, you need to calculate the gradient of the cost function for each model with respect to the parameter $\theta_j$. In other words, you need to calculate how much the cost function will change if you change $\theta j$ [51], which is called the

partial derivative, and the following equation calculates the partial derivative of the cost function with respect to the parameter $\theta j$:

$$\frac{\partial}{\partial \theta_j} MSE(\theta) = \frac{2}{m} \sum_{i=1}^{m} \left( \theta^T x^{(i)} - y^{(i)} \right) x_j^{(i)} \tag{7}$$

$MSE$ is the cost function, $\theta$ is the parameter vector, $\theta^T$ is the transpose of $\theta$, m is the number of instances, and $x^{(i)}, y^{(i)}$ is the coordinate vector of the $i$th instance.

### 3.3.2. SVM

Historical data show that SVMs are often one of the most effective methods for problem solving and are highly favored. SVMs are generalized linear classifiers that are part of supervised learning [52]. The basic principle is to determine a hyperplane by partitioning the space into multiple classes, and their decision boundary is a maximum margin hyperplane that is solved for the learned samples and can be used to partition the data non-linearly by a kernel function [53], the support vector machine formulation.

$$f(z) = \text{sign}(\sum_{i=1}^{N} \upsilon_i \Psi(z_i) + c) \tag{8}$$

Here, $\Psi$ is the mapping function that can be chosen from SVM, radial basis function (RBF), $\upsilon$ is the weight, and c is the bias.

### 3.3.3. LR

Logistic regression (LR) is often used to calculate the probability that an instance belongs to this category.

The estimated probabilities of the logistic regression model are given in Equation (9), and the output mathematical and logical values are calculated by weighing the input features.

Logistic Regression model estimated probability (vectorized form):

$$p = \sigma \left( x^T \theta \right) \tag{9}$$

x is the feature matrix and $\theta$ is the weighting factor.

$\sigma$ is a sigmoid function that outputs a number between 0 and 1. It is defined as shown in Equation (10). Logic functions:

$$\sigma(t) = \frac{1}{1 + \exp(-t)} \tag{10}$$

$\exp(-t)$ is equivalent to $e^{-t}$ and t is the independent variable.

When LR calculates the probability that x belongs to a class $p = \sigma(x^T \theta)$, then the prediction y is given as follows (11).

Logistic regression model prediction:

$$y = \begin{cases} 0, & x < 0.5 \\ 1, & x \geq 0.5 \end{cases} \tag{11}$$

y is the predicted outcome of x. Notice that $\sigma(t) < 0.5$ when $t < 0$, and $\sigma(t) \geq 0.5$ when $t \geq 0$, so a logistic regression model predicts 1 if $x^T \theta$ is positive and 0 if it is negative.

### 3.3.4. Ensemble Learning

A complete ensemble learning (EN) algorithm is roughly divided into two steps. First, the process of constructing the base learner can be parallel or serial, but serial tends to be computationally inefficient and can have an impact on subsequent base learners. The base learners are then combined, and the common combination methods used for classification

are hard and soft voting [54]. Hard voting basically works by aggregating the predictions of each base classifier and then using the result with the most votes as the final predicted class. Soft voting presupposes that all classifiers are required to be able to estimate the probability of a category, and the basic principle is to give the category with the highest average probability as predicted by averaging the probabilities over all individual classifiers. Historical data suggests that soft voting usually performs better than hard voting methods because it gives higher weights to the best classifiers [55]. Whether or not samples are put back during sampling, the sampling methods can be classified into bootstrap (bagging) and pasting. Bagging involves each sub-model randomly drawing a certain number of samples from all the sample data, putting the data back into the sample data after training is completed, and another sub-model randomly drawing the same number of sub-models from all the sample data. The bagging sampling method is favored because it can train more sub-models quickly without the limitation of the number of samples and without the dependency problem of pasting [56]. Figure 3 shows the selected GD, SVM, and LR hard and soft polling integration to generate HVG and SVG. Figure 3 below shows the selected GD, SVM, and LR hard and soft polling integration to generate HVG and SVG. Figure 4 shows the selection of RF, SVM, and LR hard and soft polling integration to generate HVG and SVG.

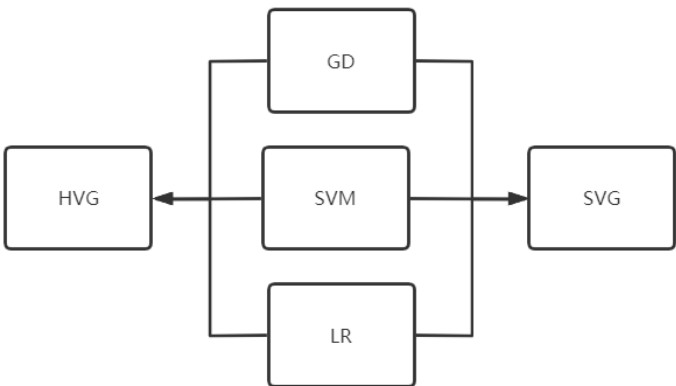

**Figure 3.** Generating hard voting HVG and soft voting SVG.

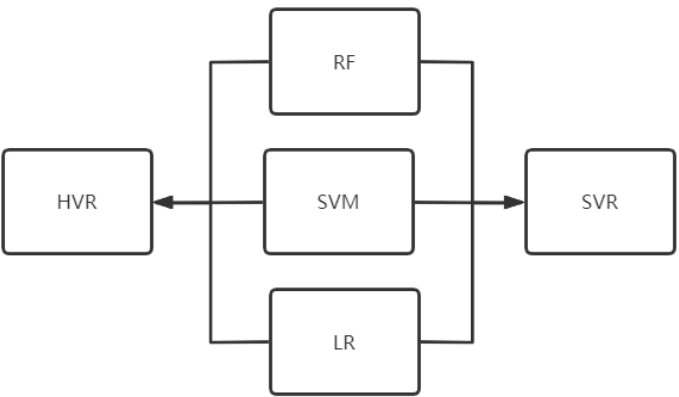

**Figure 4.** Generating hard voting HVR and soft voting SVR.

### 3.3.5. Random Forest

Random forest (RF) is based on bagging, which uses a random sampling method with put-back to draw multiple subsamples from the original dataset and uses these multiple subsamples to train multiple base learners to reduce the variance of the model. RF uses a decision tree (DT) as the base learner [57], which selects an optimal attribute division among the set of attribute features of the current tree node by means of a certain policy.

The basic principle of random forest is shown in Figure 5. The algorithm steps are as follows:

(1)   Input: sample set S = {(x,$y_{11}$), (x,$y_{22}$), . . . , (x,$y_{mm}$)}.
(2)   Output: random forest model f(x).
(3)   For t = 1, 2, . . . T (T is the number of iterations of the weak classifier).

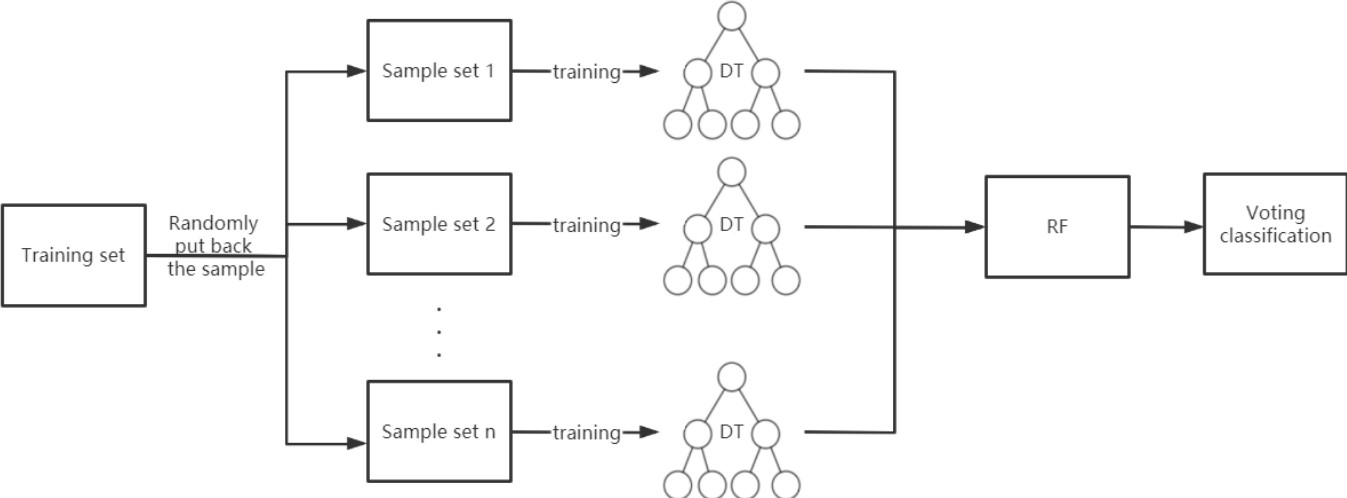

**Figure 5.** Basic principles of the random forest algorithm.

(a) The training set is randomly sampled for the tth time, and a total of m times are taken to obtain a sample set S containing m sample $S_t$.

(b) Train the tth decision tree model $G_t$ (x) with the sample set $S_t$, and when training the nodes of the decision tree model in a random selection of partial features from the sample features of the node, a calculation based on the selected partial features, and the selection of the optimal feature to do the division of left and right sub-trees, each tree will grow intact without pruning. The focus of a random forest is on "random" selection in two directions, and the randomness of these two aspects makes the random forest relatively enhanced generalization performance compared to decision trees, as they are less prone to overfitting [58].

The advantages of random forests are:

(1)   Random forest based on bagging with put-back sampling, highly parallelized training, and suitable for large data samples.
(2)   Random sampling is used, the model has low variance. It has a strong resistance to noise and a strong generalization ability.
(3)   Due to the random selection of the features to be selected, good training predictions can also be achieved for high-dimensional features.
(4)   Relative to the boosting algorithm [59], the implementation is relatively simple, the accuracy is high, and the training is fast.

The basic principle of the model selection algorithm is shown in Figure 6. The model selection algorithm is based on the feature selection algorithm, and the aim is to use the 21 features selected by the feature selection algorithm in the previous section to initially select algorithmic models with high Accuracy, Precision, Recall, F1_Score, strong generalization ability, and short prediction time using the model selection algorithm. First, the resultant features ENF from the feature selection algorithm were trained and predicted using GD, SVM, LR, RF, HVG, SVG, HVR, and SVR, respectively, and then, the performance of each model was evaluated using the model evaluation metrics, Accuracy, Precision, Recall, F1_Score, and the average value, Average, was combined with the prediction_time generated by each model to select a model with high accuracy and short prediction time for DDoS detection. The model algorithm pseudo-code is shown in Algorithm 2: model selection algorithm.

| **Algorithm 2**: Model Selection Algorithm |
|---|
| **Input:** feature_selection_results, GD, SVM, LR, RF, HVG, SVG, HVR, SVR |
| **Output:** Select 1 most important Model. |
| **procedure:** Top Model |

**Input:** feature_selection_results, GD, SVM, LR, RF, HVG, SVG, HVR, SVR
**Output:** Select 1 most important Model.
**procedure:** Top Model

    Step 1: Generate the dataset after Feature Selection from the output of Algorithm 1 FS_dataset

    Step 2: Applying FS_dataset, generate the GD confusion matrix and calculate the corresponding Average

    Step 3: defined variable selected_models = [ ]

    Step 4: dataset = [CIC-IDS2018, CIC-IDS2017, CIC-DoS2016]

    Step 5: models = [GD, SVM, LR, RF, HVG, SVG,HVR, SVR]

    Step 6: for each ds in dataset:

      for each ms in models:

               ms. Average # Calculate Average for each model

               selected_models.append(ms.f1_score)

    Step 7: defined variable model_selection_results = [ ]

    Step 8: for each model_Average in model_selection_results:

             model_selection_result = model_selection_result.index(0) #initialize

             If(model_Average > model_selection_result):

                 model_selection_result = model_f1_score

    Step 9: model_selection_result

**end procedure**

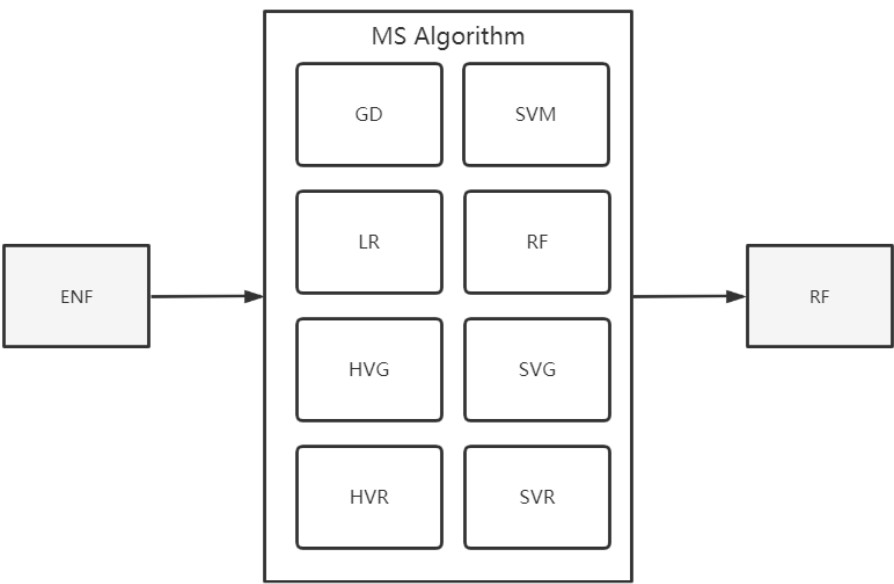

**Figure 6.** Basic principles of the model selection algorithm.

*3.4. RF Optimization Algorithms*

In the data network, due to the large amount of data traffic, data imbalance, and various types of data, the detection algorithm for DDoS attacks needs to be highly parallelized, with simple models, strong generalization capabilities, and high detection efficiency. There is no doubt that a random forest-based DDoS attack detection algorithm can take on this task. When the maximum depth of the random forest and the number of decision trees are too large, the time complexity and space complexity of training and detection will be higher, so some optimization is needed. The main parameters of the random forest to be optimized are as follows.

(a)    max_samples: The training set size is set by max_samples.

(b)    max_depth: The maximum depth of the decision tree, the default is not limited, if the model has a lot of samples and features, it is recommended to limit this. The common value can be between 10 and 100. If the model has a large number of samples and

features, it is recommended to modify this limit, which can commonly be taken to be between 10 and 100.

(c) n_estimators: Specifies the number of decision trees in the random forest, default is 100. n_estimators is too small to be easily under-fitted and too large to be computationally intensive, so the parameters need to be optimized to a moderate value. The pseudo-code for the optimization algorithm is given in Algorithm 3: random forest optimization algorithm.

---

**Algorithm 3**: Random Forest Optimization Algorithm

---

**Input:** FS_dataset, max_samples, max_depth, n_estimators
**Output:** Random Forest Optimization Model
**procedure:** parameter Optimization
    Step 1: defined variable results = [Method,matrix, Accuracy, Precision,
                                    Recall, F1_Score, Average, predict_time]
    Step 2: Initialize the parameter RandomForestClassifier(max_samples = 0.9,
                                      max_depth = 20, n_estimators = 100)
    Step 3: Optimising max_samples
        for sam in [0.1,0.2,0.3,0.4,0.5,0.6,0.7,0.8,0.9]:
      model = RandomForestClassifier(max_samples = sam,
                                max_depth = 20, n_estimators = 100)
      model.fit
        testes_out(y_test,y_pred) # Performance test function
           results.append(Method, matrix, Accuracy, Precision, Recall,
                    F1_Score, Average,predict_time)
    Step 4: Select the opt_max_samples with the highest results
    Step 5: Optimising max_depth
        for dep in range(10,30,2):
        model = RandomForestClassifier(max_samples = 0.9,
                                max_depth = dep, n_estimators = 100)
        model.fit
        testes_out(y_test,y_pred) # Performance test function
        results.append(Method, matrix, Accuracy, Precision, Recall,
                    F1_Score, Average,predict_time)

    Step 6: Select the opt_max_depth with the highest results
    Step 7: Optimising n_estimators
        for dep in range(10,210,20):
        model = RandomForestClassifier(max_samples = 0.9,
                                max_depth = 20, n_estimators = est)
        model.fit
        testes_out(y_test,y_pred) # Performance test function
        results.append(Method, matrix, Accuracy, Precision, Recall,
                    F1_Score, Average,predict_time)
    Step 8: Select the opt_n_estimators with the highest results
    Step 9: #Output the optimization result:
    RandomForestClassifier(max_samples = opt_max_samples,
                          max_depth = opt_max_depth,
                        n_estimators = opt_n_estimators)
**end procedure**

---

### 3.5. Performance Metrics and Model Evaluation

This experiment tests the performance results of the experimental study based on a confusion matrix to evaluate the experimental results. The confusion matrix contains both estimated and actual values. The confusion matrix is shown in Figure 7. Here, true positive TP and true negative TN indicate correct predicted values, while false positive FP and false negative FN indicate incorrect predicted values [60]. In addition, model performance was assessed using the subject operating curve ROC and the area under the curve AUC. The ROC curve has a false positive rate on the horizontal axis and a true positive rate on the vertical axis.

| Predicted<br>Observed | Postive | Negtive | count |
|---|---|---|---|
| Positive | True Positive (TP) | False Negtive (FN) | P |
| Negtive | False Positive (FP) | True Negtive (TN) | N |

**Figure 7.** Confusion matrix.

The confusion matrix contains information on the evaluation of the lower right true negative, lower left false positive, upper right false negative, and upper left true. The proposed model is evaluated according to the performance metrics from the confusion matrix and the formulas Accuracy (12), Precision (13), Recall (13), F1_Score (14), and Average. The formulas for these metrics are given in the following equation:

$$\text{Accuracy} = \frac{\text{TP} + \text{TN}}{\text{TP} + \text{TN} + \text{FP} + \text{FN}} \tag{12}$$

$$\text{Precision} = \frac{\text{TP}}{\text{TP} + \text{FP}} \tag{13}$$

$$\text{Recall} = \frac{\text{TP}}{\text{TP} + \text{FN}} \tag{14}$$

This experiment also used a 3-fold cross-validation method to determine the error of the model. In order to conduct the experimental study, the dataset was used over 100,000 items, including 21 different DDoS attack features. f1_Score formula as in (15), when compared to a single comparison of different classifiers Precision or Recall with bias, F1_Score combines Precision and Recall, and the classifier will only have a higher F1 score when both Precision and Recall are high, and so, the sub-classifier will be better.

$$\text{F}_1 = \frac{2}{\frac{1}{\text{Precision}} + \frac{1}{\text{Recall}}} = \frac{2 * TP}{2 * TP + FP + FN} \tag{15}$$

The AOC of a perfect classifier should be as shown in Figure 8 below, which corresponds to an AUC of 1. The dashed line indicates the ROC curve of a purely random classifier, which corresponds to an AUC of 0.5.

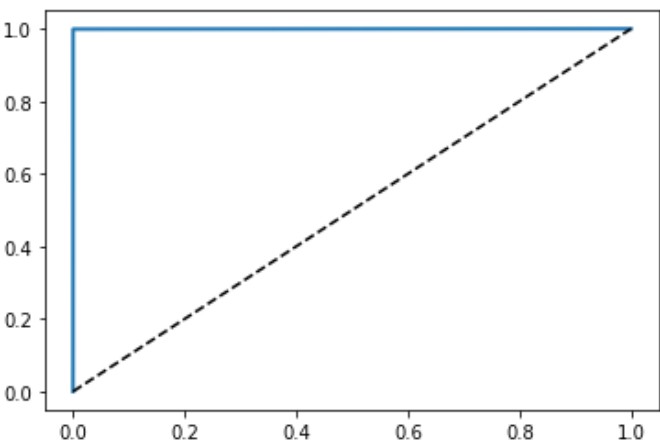

**Figure 8.** AOC of good classifiers.

## 4. Experimental Results

### 4.1. Feature Selection

The 79 features obtained from feature extraction of each dataset after pre-processing from CIC-IDS2018, CIC-IDS2017, and CIC-DoS2016 synthesized DDoS and CIC-DDoS2019 were coded in Table 1 below and used as inputs to the feature selection algorithm.

**Table 1.** Dataset characteristics and their coding.

| Id | Feature Name | Id | Feature Name | Id | Feature Name | Id | Feature Name |
|---|---|---|---|---|---|---|---|
| 1 | Source Port | 21 | Flow IAT Max | 41 | Min Packet Length | 61 | Bwd Avg Bytes/Bulk |
| 2 | Destination Port | 22 | Flow IAT Min | 42 | Max Packet Length | 62 | Bwd Avg Packets/Bulk |
| 3 | Protocol | 23 | Fwd IAT Total | 43 | Packet Length Mean | 63 | Bwd Avg Bulk Rate |
| 4 | Flow Duration | 24 | Fwd IAT Mean | 44 | Packet Length Std | 64 | Subflow Fwd Packets |
| 5 | Total Fwd Packets | 25 | Fwd IAT Std | 45 | Packet Length Variance | 65 | Subflow Fwd Bytes |
| 6 | Total Backward Packets | 26 | Fwd IAT Max | 46 | FIN Flag Count | 66 | Subflow Bwd Packets |
| 7 | Total Length of Fwd Packets | 27 | Fwd IAT Min | 47 | SYN Flag Count | 67 | Subflow Bwd Bytes |
| 8 | Total Length of Bwd Packets | 28 | Bwd IAT Total | 48 | RST Flag Count | 68 | Init_Win_bytes_forward |
| 9 | Fwd Packet Length Max | 29 | Bwd IAT Mean | 49 | PSH Flag Count | 69 | Init_Win_bytes_backward |
| 10 | Fwd Packet Length Min | 30 | Bwd IAT Std | 50 | ACK Flag Count | 70 | act_data_pkt_fwd |
| 11 | Fwd Packet Length Mean | 31 | Bwd IAT Max | 51 | URG Flag Count | 71 | min_seg_size_forward |
| 12 | Fwd Packet Length Std | 32 | Bwd IAT Min | 52 | CWE Flag Count | 72 | Active Mean |
| 13 | Bwd Packet Length Max | 33 | Fwd PSH Flags | 53 | ECE Flag Count | 73 | Active Std |
| 14 | Bwd Packet Length Min | 34 | Bwd PSH Flags | 54 | Down/Up Ratio | 74 | Active Max |
| 15 | Bwd Packet Length Mean | 35 | Fwd URG Flags | 55 | Average Packet Size | 75 | Active Min |
| 16 | Bwd Packet Length Std | 36 | Bwd URG Flags | 56 | Avg Fwd Segment Size | 76 | Idle Mean |
| 17 | Flow Bytes/s | 37 | Fwd Header Length | 57 | Avg Bwd Segment Size | 77 | Idle Std |
| 18 | Flow Packets/s | 38 | Bwd Header Length | 58 | Fwd Avg Bytes/Bulk | 78 | Idle Max |
| 19 | Flow IAT Mean | 39 | Fwd Packets/s | 59 | Fwd Avg Packets/Bulk | 79 | Idle Min |
| 20 | Flow IAT Std | 40 | Bwd Packets/s | 60 | Fwd Avg Bulk Rate | | |

Table 2 below shows the feature selection algorithm's processing of variance, mutual information from filter, backward elimination from wrapper and Lasso.L1, and random forest from embedded for the feature selection method of the top 25 features selected.

**Table 2.** Feature selection algorithm process.

| Feature Selection Method | Id |
|---|---|
| filter-Variance method | 1, 2, 3, 4, 8, 9, 11, 12, 17, 21, 26, 37, 38, 42, 43, 45, 50, 55, 56, 67, 68, 69, 71 |
| wrapper-Backward Elimination | 1, 2, 3, 8, 11, 17, 21, 25, 26, 28, 31, 32, 33, 34, 41, 45, 50, 51, 55, 66, 67, 68, 69 |
| embed-Lasso.l1 | 1, 2, 3, 4, 8, 9, 11, 12, 17, 21, 26, 37, 38, 42, 43, 45, 50, 55, 56, 67, 68, 69, 71 |
| filter-Mutual Information | 2, 7, 8, 9, 11, 12, 13, 15, 16, 21, 22, 37, 38, 42, 43, 44, 45, 55, 56, 65, 67, 69, 71 |
| embed-Random Forest | 2, 3, 6, 9, 11, 12, 14, 17, 18, 19, 21, 26, 37, 38, 39, 42, 43, 50, 56, 67, 68, 69, 71 |
| 5 Select 4 results | 2, 3, 8, 9, 11, 12, 17, 21, 26, 37, 38, 42, 43, 45, 50, 55, 56, 67, 68, 69, 71 |

The final feature results obtained by the feature selection algorithm. The machine features are coded in Table 3.

**Table 3.** Feature selection algorithm processing results.

| Id | Feature Name | Id | Feature Name |
|----|-------------|----|-------------|
| 1 | Destination Port | 12 | Fwd Header Length |
| 2 | Fwd Packet Length Mean | 13 | Bwd Header Length |
| 3 | Flow IAT Max | 14 | Max Packet Length |
| 4 | Subflow Bwd Bytes | 15 | Packet Length Mean |
| 5 | Init_Win_bytes_backward | 16 | Packet Length Variance |
| 6 | Protocol | 17 | ACK Flag Count |
| 7 | Total Length of Bwd Packets | 18 | Average Packet Size |
| 8 | Fwd Packet Length Max | 19 | Avg Fwd Segment Size |
| 9 | Fwd Packet Length Std | 20 | Init_Win_bytes_forward |
| 10 | Flow Bytes/s | 21 | min_seg_size_forward |
| 11 | Fwd IAT Max | | |

### 4.2. Model Selection

We chose GD, SVM, LR, RF, HVG, SVG, HVR, and SVR according to the model selection algorithm, and the experimental results are shown in Table 4 below.

**Table 4.** Performance indicators for each model of the model selection algorithm.

| Method | Matrix | | Accuracy | Precision | Recall | F1_Score | Average | Normal_Detect_Rate | Atk_Detect_Rate | Predict_Time |
|--------|--------|--------|----------|-----------|--------|----------|---------|---------------------|------------------|--------------|
| GD | 15,275<br>1506 | 858<br>15,361 | 0.928364 | 0.947099 | 0.910713 | 0.928550 | 0.928681 | 0.946817 | 0.910713 | 0.000989 |
| SVM | 15,343<br>1514 | 790<br>15,353 | 0.930182 | 0.951062 | 0.910239 | 0.930203 | 0.930422 | 0.951032 | 0.910239 | 5.717282 |
| LR | 15,206<br>1514 | 927<br>15,353 | 0.926030 | 0.943059 | 0.910239 | 0.926358 | 0.926422 | 0.942540 | 0.910239 | 0.002003 |
| RF | 16,119<br>8 | 14<br>16,859 | 0.999333 | 0.999170 | 0.999526 | 0.999348 | 0.999344 | 0.999132 | 0.999526 | 0.216050 |
| HVG | 15,298<br>1511 | 835<br>15,356 | 0.928909 | 0.948428 | 0.910417 | 0.929034 | 0.929197 | 0.948243 | 0.910417 | 5.845312 |
| SVG | 15,498<br>1510 | 635<br>15,357 | 0.935000 | 0.960293 | 0.910476 | 0.934721 | 0.935122 | 0.960640 | 0.910476 | 11.845665 |
| HVR | 15,921<br>120 | 212<br>16,747 | 0.989939 | 0.987499 | 0.992886 | 0.990185 | 0.990127 | 0.986859 | 0.992886 | 12.066660 |
| SVR | 15,926<br>76 | 207<br>16,791 | 0.991424 | 0.987822 | 0.995494 | 0.991643 | 0.991596 | 0.987169 | 0.995494 | 12.069690 |

First, we analyzed the Accuracy, Precision, Recall and F1_Score of each method separately, based on the generated confusion matrix. As shown in Figure 9, most of the tested methods achieved good results, with even the worst LR achieving 92% accuracy, while RF, HVR, and SVR had particularly good results for the metrics. In terms of individual algorithmic models, there is no doubt that the RF method ranked first in all categories, with an overwhelming advantage in terms of Accuracy, Precision, Recall and F1_Score, which were all close to 100%, while taking only 0.2 s in prediction time. In terms of the integrated models, soft voting tended to perform better than hard voting.

The integrated combination of SVM, LR, and RF had a better performance than the integrated combination of GD, SVM, and LR, indicating that the way the models are combined can have a critical impact on the results. RF had better performance than HVR and SVR on all items. It shows that the base learner is not necessarily weaker than the integrated learning when the prediction is better. The contrast shows that RF has the best performance compared to other models in all performance metrics, so the model with the better results in the algorithm stage should be RF.

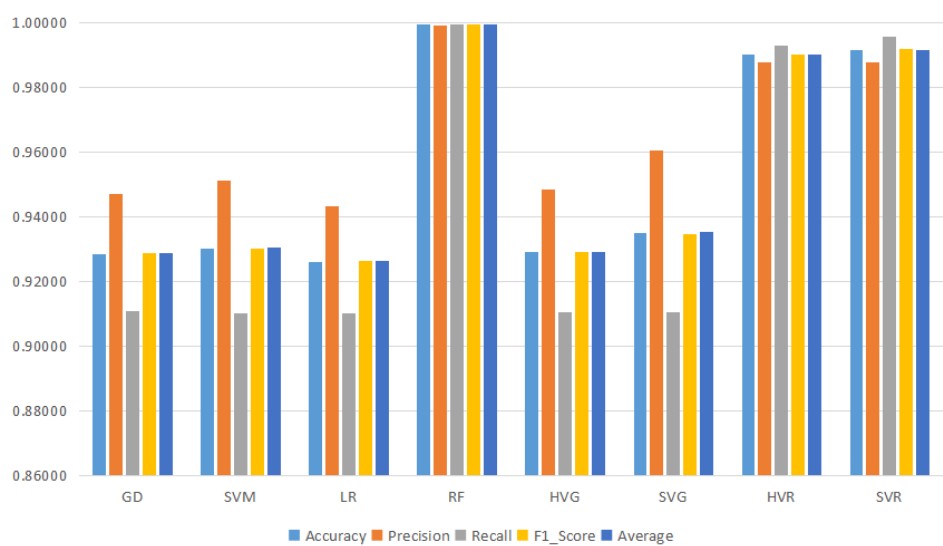

**Figure 9.** Performance of each model.

AUC is defined as the area under the ROC curve enclosed by the coordinate axis. As shown in Figure 10, The ROC curve of each model shows that the ROC curve of RF is the closest to the standard ROC curve Figure 11, and the AUC scores of each model are shown in Table 5, where the AUC score of RF is 0.99948, which is the best performance among the models, corresponding to the histogram Figure 12.

**Figure 10.** ROC curves of GD, SVM, LR, RF, HVG, SVG, HVR, SVR.

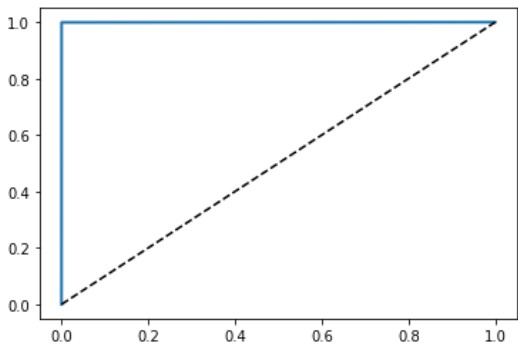

**Figure 11.** Standard ROC curve.

**Table 5.** AUC scores for model selection algorithms GD, SVM, LR, RF, HVG, SVG, HVR, and SVR.

|     | GD | SVM | LR | RF | HVG | SVG | HVR | SVR |
|-----|-----|-----|-----|-----|-----|-----|-----|-----|
| AUC | 0.9292 | 0.9306 | 0.9265 | 0.9995 | 0.9291 | 0.9414 | 0.9894 | 0.9912 |

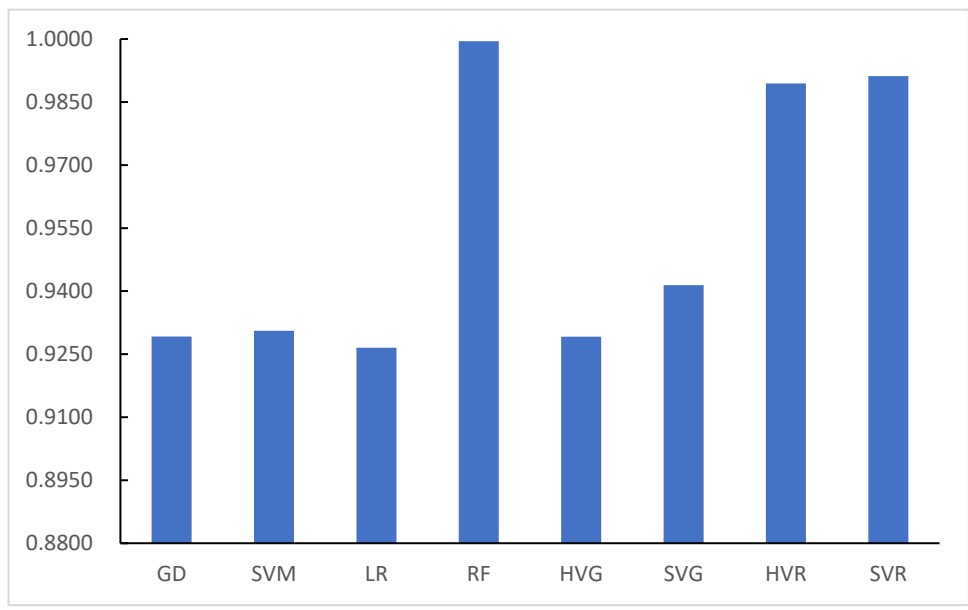

**Figure 12.** AUC score for model selection algorithms GD, SVM, LR, RF, HVG, SVG, HVR, and SVR.

Model selection algorithm AUC scores for each model.

### 4.3. RF Optimization

After the initial selection of a suitable model algorithm for RF in the model selection algorithm phase, the initially selected RF parameters: random forest sampling rate, maximum depth per tree, and number of trees were then optimized by the RF optimization algorithm in order to further improve its DDoS detection performance. Firstly, we set max_samples = 0.9, max_depth = 20 and n_estimators = 100 to initialize the parameters of the RF before optimization. The resulting Accuracy, Precision, Recall, F1_Score, Average, and predict_time are shown in Table 6 below.

**Table 6.** Performance indicators before RF optimization.

|         | Accuracy | Precision | Recall | F1_Score | Average | Predict_Time |
|---------|----------|-----------|--------|----------|---------|--------------|
| bef_opt | 0.9993   | 0.9992    | 0.9993 | 0.9993   | 0.9993  | 0.2164       |

### 4.3.1. Exploring Maximum Sampling

The RF optimization algorithm performs the optimization of the parameter max_samples, in order to control the variables, always keeping the parameter max_depth = 20 and n_estimators = 100, and max_samples is selected for the parameter interval [0.1, 0.9], with each incremental step being 0.1.

The results obtained are shown in Table 7 below.

**Table 7.** max_samples optimization performance table.

| Max_Samples | Accuracy | Precision | Recall | F1_Score | Average | Predict_Time |
|---|---|---|---|---|---|---|
| 0.1 | 0.9986 | 0.9985 | 0.9987 | 0.9986 | 0.9986 | 0.1999 |
| 0.2 | 0.9989 | 0.9990 | 0.9989 | 0.9990 | 0.9990 | 0.2043 |
| 0.3 | 0.9991 | 0.9991 | 0.9991 | 0.9991 | 0.9991 | 0.2050 |
| 0.4 | 0.9992 | 0.9992 | 0.9992 | 0.9992 | 0.9992 | 0.2031 |
| 0.5 | 0.9993 | 0.9993 | 0.9993 | 0.9993 | 0.9993 | 0.2161 |
| 0.6 | 0.9993 | 0.9992 | 0.9995 | 0.9993 | 0.9993 | 0.2098 |
| 0.7 | 0.9994 | 0.9992 | 0.9995 | 0.9994 | 0.9994 | 0.2056 |
| 0.8 | 0.9992 | 0.9992 | 0.9993 | 0.9993 | 0.9993 | 0.2121 |
| 0.9 | 0.9994 | 0.9994 | 0.9994 | 0.9994 | 0.9994 | 0.2114 |

To facilitate the analysis, we selected three important performance metrics from Table 7, namely Accuracy, F1_Score, and Average, and made the corresponding line graphs, Figure 13: As can be seen from the graphs, the trends and overlap of Accuracy, F1_Score, and Average are very consistent. It can be seen that as the maximum sampling max_samples increases, Accuracy, F1_Score, and Average all show a corresponding increase, but at max_samples = 0.5, the growth slows down significantly, and at max_samples = 0.9, Accuracy, F1_Score, and Average achieved the maximum value when max_samples = 0.9.

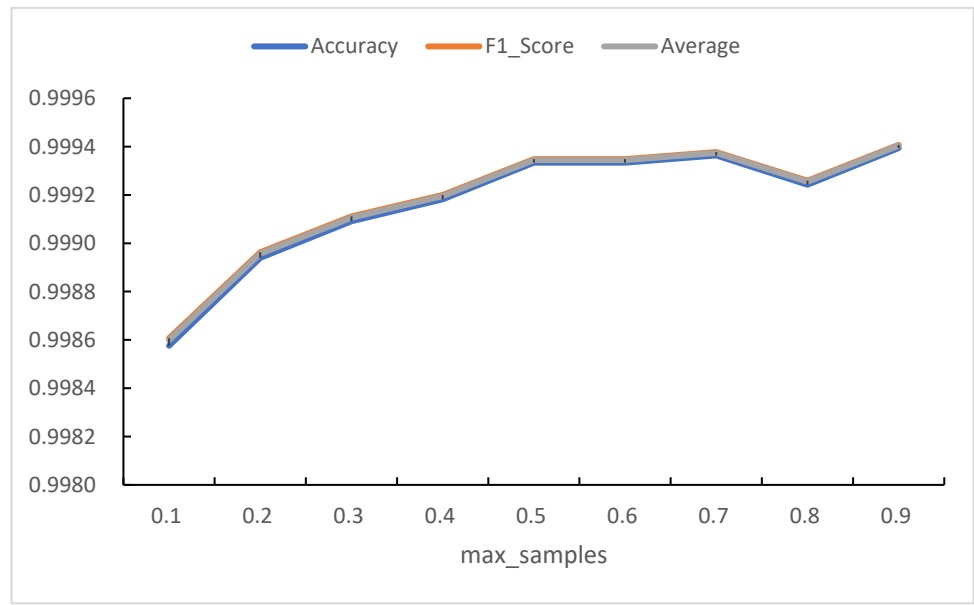

**Figure 13.** max_samples optimizes Accuracy, F1_Score, and Average.

To facilitate the analysis of the concern between max_samples and model prediction time, we chose the predict_time indicator from Table 7 for the following Figure 14, from which we see that predict_time shows a wave-like pattern, with max_samples. However, the increasing size of the predict_time does not show a significant increase, indicating that the increase in the maximum sampling and the impact on the prediction time is not significant.

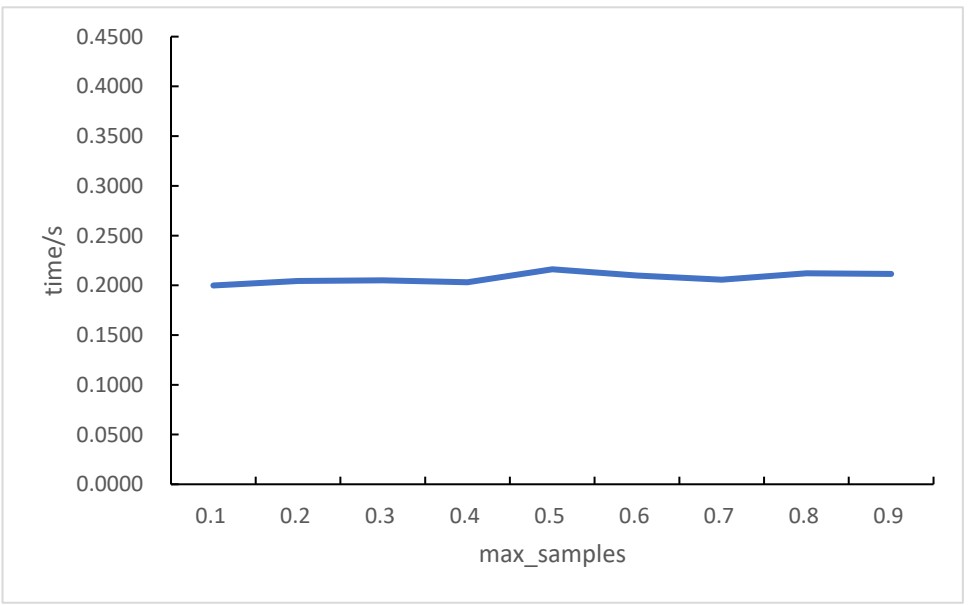

**Figure 14.** max_samples optimizes predict_time.

In summary, combining the Method-Accuracy, F1_Score, and Average line graphs, since the best results for each metric are achieved at max_samples = 0.9, and the Method-predict_time line graph illustrates that an increase in maximum samples has little effect on prediction time, we can conclude that the best results are achieved when we set the parameter max_samples samples to 0.9.

### 4.3.2. Exploring the Maximum Depth

The RF optimization algorithm always keeps the parameter max_samples = 0.9 when performing the optimization of the parameter max_depth in order to control the variables, n_estimators = 100, max_depth selects a parameter interval in the range [10, 28], and each incremental step is 2. The results obtained are shown in Table 8 below.

**Table 8.** max_depth optimization performance table.

| Max_Depth | Accuracy | Precision | Recall | F1_Score | Average | Predict_Time |
|:---:|:---:|:---:|:---:|:---:|:---:|:---:|
| 10 | 0.9985 | 0.9977 | 0.9993 | 0.9985 | 0.9985 | 0.1870 |
| 12 | 0.9987 | 0.9981 | 0.9994 | 0.9988 | 0.9987 | 0.1945 |
| 14 | 0.9990 | 0.9984 | 0.9996 | 0.9990 | 0.9990 | 0.1992 |
| 16 | 0.9993 | 0.9991 | 0.9995 | 0.9993 | 0.9993 | 0.2039 |
| 18 | 0.9992 | 0.9992 | 0.9993 | 0.9992 | 0.9992 | 0.2063 |
| 20 | 0.9993 | 0.9992 | 0.9994 | 0.9993 | 0.9993 | 0.2106 |
| 22 | 0.9993 | 0.9993 | 0.9993 | 0.9993 | 0.9993 | 0.2102 |
| 24 | 0.9993 | 0.9993 | 0.9994 | 0.9993 | 0.9993 | 0.2100 |
| 26 | 0.9993 | 0.9993 | 0.9994 | 0.9993 | 0.9993 | 0.2142 |
| 28 | 0.9994 | 0.9993 | 0.9994 | 0.9994 | 0.9994 | 0.2091 |

Similarly, we selected three important performance indicators from Table 8, namely Accuracy, F1_Score, and Average, and made the corresponding Method-Accuracy, F1_Score, and Average line graphs in Figure 15: As can be seen from the graphs, the trend direction and overlap of the three curves of Accuracy, F1_Score, and Average are also very consistent, from which we can see that as the maximum sampling, max_depth keeps increasing, Accuracy, F1_Score, and Average, which show a corresponding increase, and a turning point occurs when max_depth = 16. When max_depth continues to increase, Accuracy, F1_Score, and Average do not increase significantly when max_depth = 18, and they even decrease somewhat when max_depth = 18. The difference between the performance

indicators after max_depth = 20 and those at max_depth = 16 is not significant. The graph also reflects that blindly increasing the max_depth may not improve the performance of the model, but may bring additional overhead to the model and affect the other performance of the model.

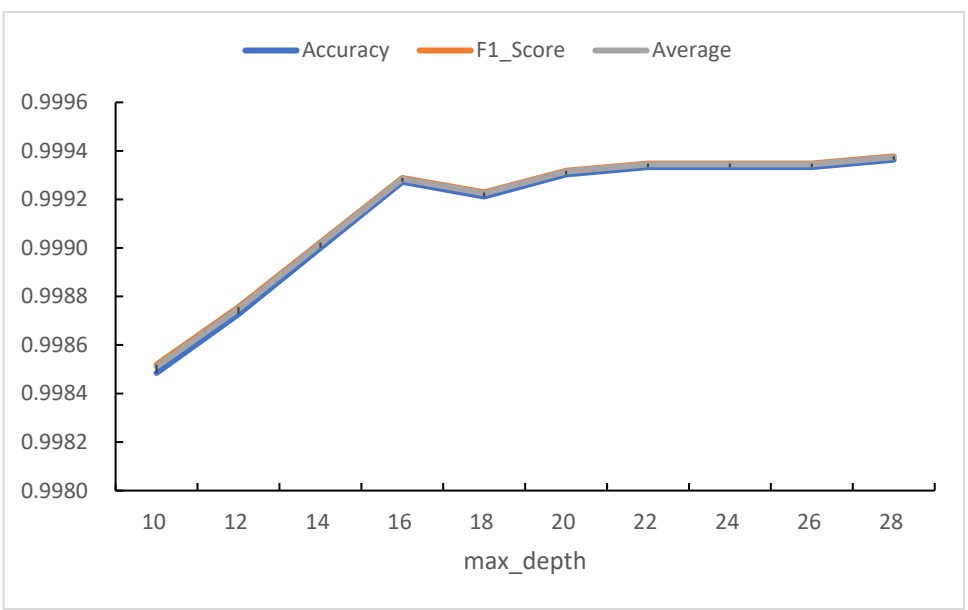

**Figure 15.** max_depth optimizes Accuracy, F1_Score, and Average.

Similarly, to facilitate the analysis of the relationship between max_depth and the model prediction time predict_time, we take the data from Table 8, which is plotted in Figure 16. The predict_time does not increase as the max_depth increases, and even after max_depth = 20, there is an up and down variation. It can be seen that max_depth and predict_time have a great influence on the prediction time within a certain range, but the influence is not constant.

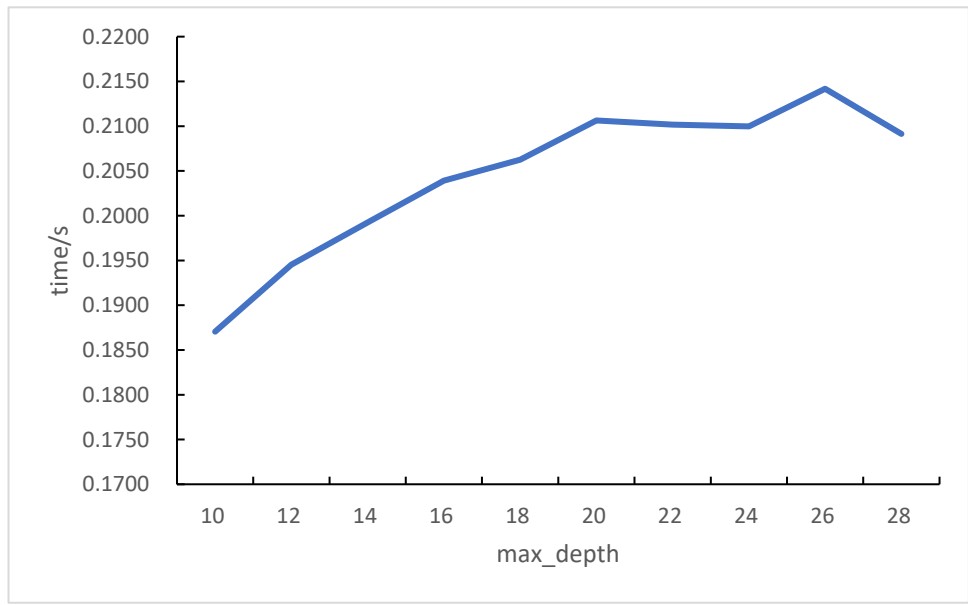

**Figure 16.** max_depth optimization predict_time.

From a comprehensive point of view, from the Method-Accuracy, F1_Score, and Average line graphs, it can be seen that within the maximum depth max_depth of 20, the

prediction time predict_time is greatly influenced by max_depth, while in the Method-Accuracy, F1_Score, and Average, the Accuracy, F1_Score, and Average indicators have reached a more ideal state when max_depth = 16 in the line graph, and so, the optimization result of this parameter max_depth is 16.

### 4.3.3. Exploring the Decision Trees

The RF optimization algorithm was used to optimize the n_estimators of the parametric decision tree, keeping the parameters max_samples = 0.9 and max_depth = 20 in order to control the variables. The n_estimators were selected in the parameter interval range [10, 190], with each incremental step being 20. The results are shown in Table 9 below.

**Table 9.** n_estimators optimization performance table.

| n_Estimators | Accuracy | Precision | Recall | F1_Score | Average | Predict_Time |
|---|---|---|---|---|---|---|
| 10 | 0.9992 | 0.9991 | 0.9994 | 0.9993 | 0.9993 | 0.0241 |
| 30 | 0.9993 | 0.9993 | 0.9993 | 0.9993 | 0.9993 | 0.0671 |
| 50 | 0.9994 | 0.9993 | 0.9996 | 0.9994 | 0.9994 | 0.1043 |
| 70 | 0.9993 | 0.9992 | 0.9994 | 0.9993 | 0.9993 | 0.1492 |
| 90 | 0.9994 | 0.9993 | 0.9995 | 0.9994 | 0.9994 | 0.1900 |
| 110 | 0.9993 | 0.9993 | 0.9993 | 0.9993 | 0.9993 | 0.2283 |
| 130 | 0.9993 | 0.9992 | 0.9994 | 0.9993 | 0.9993 | 0.2703 |
| 150 | 0.9993 | 0.9992 | 0.9994 | 0.9993 | 0.9993 | 0.3146 |
| 170 | 0.9993 | 0.9993 | 0.9994 | 0.9993 | 0.9993 | 0.3542 |
| 190 | 0.9993 | 0.9993 | 0.9994 | 0.9993 | 0.9993 | 0.3946 |

Similarly, we selected three indicators from Table 9, namely, Accuracy, F1_Score, and Average, which are important performance indicators, and made the corresponding Method-Accuracy, F1_Score, and Average line graphs Figure 17. As can be seen from the graphs, the trend direction and overlap of the three curves of Accuracy, F1_Score, and Average are basically the same, from which we can see that in the n_estimators interval [10, 50], as the n_estimators of the decision tree increases, Accuracy, F1_Score, and Average keep increasing, and when the decision tree is 50, Accuracy, F1 _Score, and Average peaked when the decision tree was 50, and then the indicators showed fluctuations with the increase in the decision tree, and finally leveled off. It also shows that too large a decision tree does not improve performance, but may bring additional overhead to the model and increase the prediction time of the model.

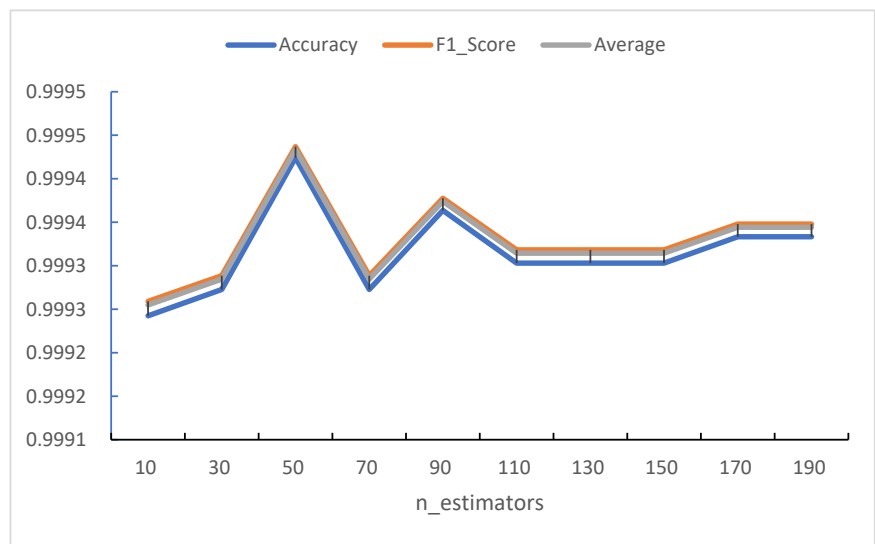

**Figure 17.** n_estimators optimize Accuracy, F1_Score, and Average.

Similarly, to facilitate the analysis of the relationship between the decision tree and the model prediction time predict_time, we selected the predict_time metric from Table 9 and plotted it as Figure 18 below. The relationship between predict_time and max_depth increases linearly.

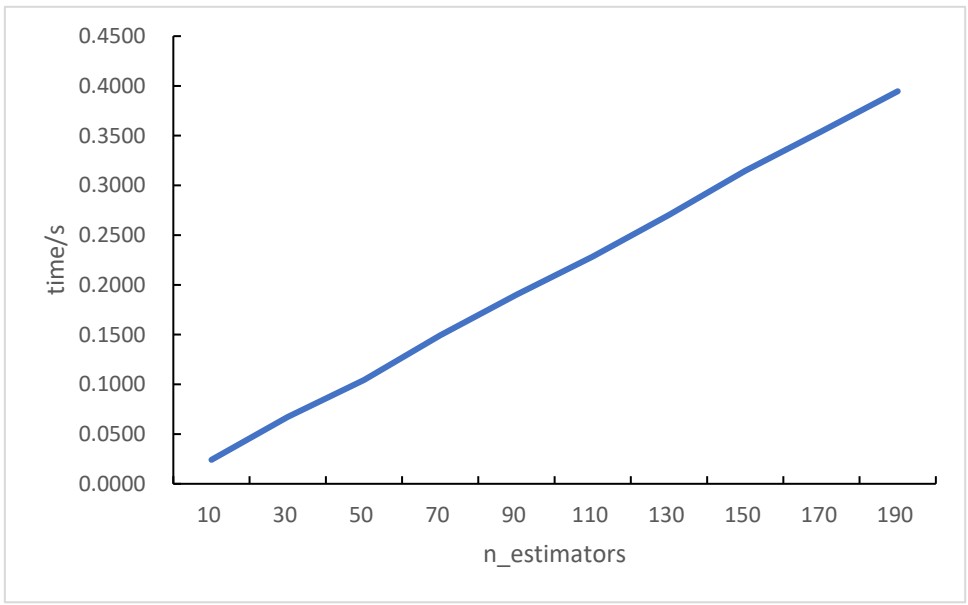

**Figure 18.** n_estimators optimizes predict_time.

In summary, the prediction time in the n_estimators-predict_time line graph shows a linear increase in prediction time and max_depth. In the Method-Accuracy, F1_Score, and Average line graphs, the performance of each tree reaches the maximum point when the decision tree is 50, so the optimization result for n_estimators is 50.

### 4.3.4. Optimization Results

The new model parameters and performance test results after the RF optimization algorithm are shown in Table 10 and Figure 19, corresponding to the confusion matrix and ROC as Figures 20 and 21, respectively. The model = RandomForestClassifier (max_samples = 0.9, max_depth = 16, n_estimators = 50).

For the Accuracy, Precision, Recall, F1_Score, and Average, each metric is close to 100% and the predict_time is only 0.1 s, which also facilitates the subsequent deployment of the model to the production environment for real-time testing.

The comparison between before and after optimization is shown in Table 11, Figures 22 and 23 below. Accuracy increased by 0.00006, Recall increased by 0.00018, and Average increased by 0.00006; however, the prediction time was reduced from 0.21636 to 0.10816, which is 0.10820 less than the original, reducing the prediction time by more than half. This is certainly significant in a system with high real-time requirements, and provides important support for the model to be deployed in production for real-time DDoS detection, and also shows important practicality.

**Table 10.** RF optimized performance indicators.

|  | Accuracy | Precision | Recall | F1_Score | Average | Predict_Time |
|---|---|---|---|---|---|---|
| aft_opt | 0.9993 | 0.9992 | 0.9995 | 0.9993 | 0.9993 | 0.1082 |

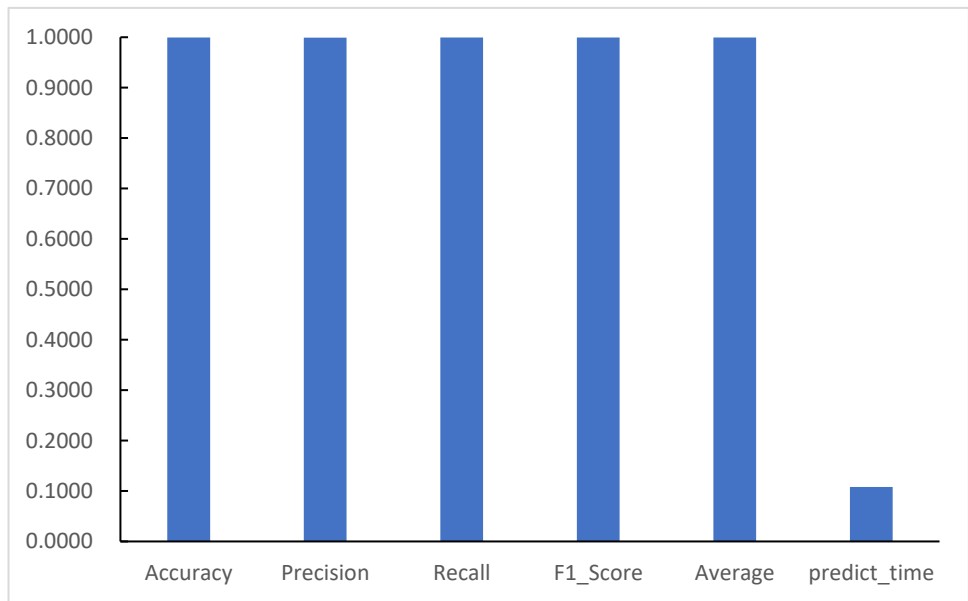

**Figure 19.** RF optimized performance metrics chart.

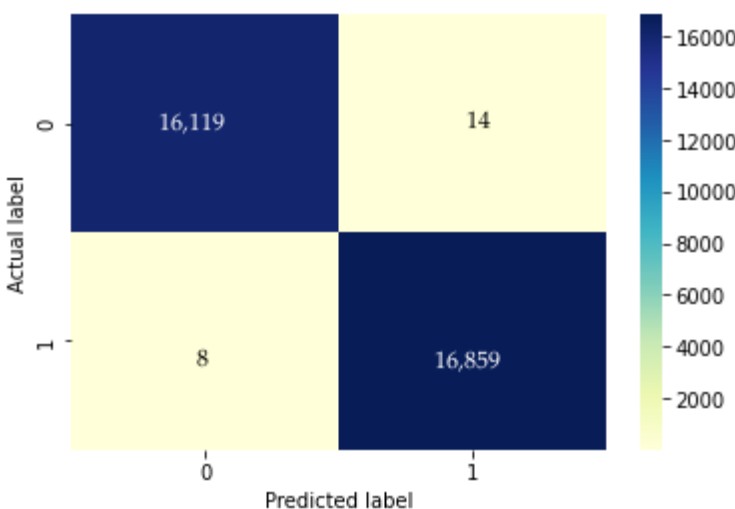

**Figure 20.** Confusion matrix after RF optimization.

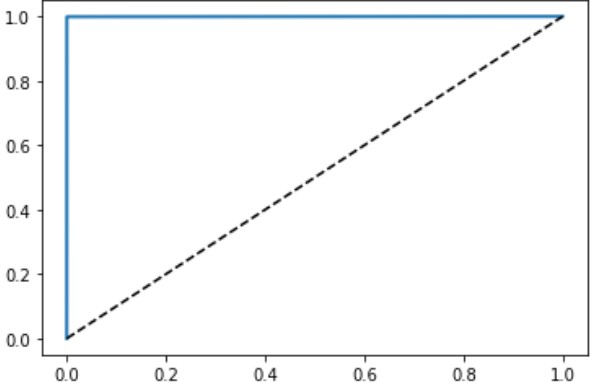

**Figure 21.** ROC after RF optimization.

**Table 11.** Comparison of performance indicators before and after RF optimization.

|  | **Accuracy** | **Precision** | **Recall** | **F1_Score** | **Average** | **Predict_Time** |
|---|---|---|---|---|---|---|
| bet_opt | 0.9993 | 0.9992 | 0.9993 | 0.9993 | 0.9993 | 0.2164 |
| aft_opt | 0.9993 | 0.9992 | 0.9995 | 0.9993 | 0.9993 | 0.1082 |
| Difference | 0.0001 | −0.0001 | 0.0002 | 0.0001 | 0.0001 | −0.1082 |

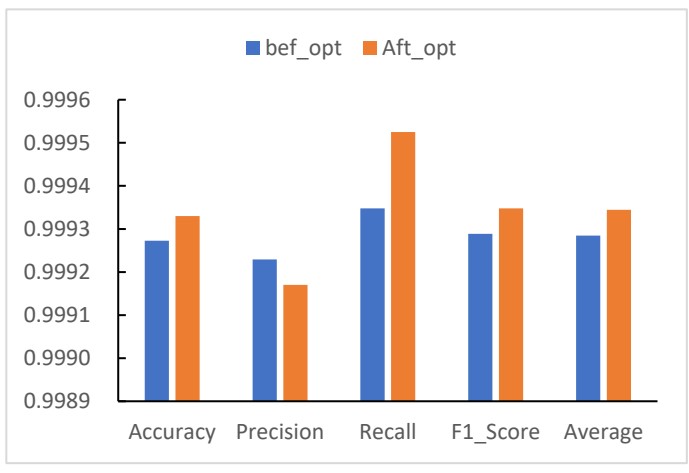

**Figure 22.** Performance comparison before and after RF optimization.

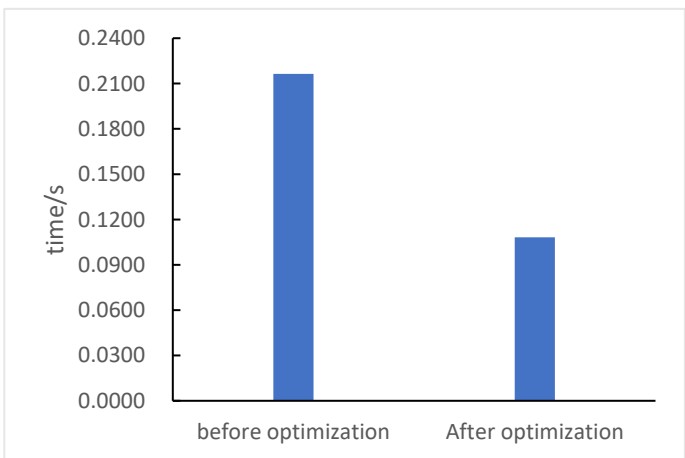

**Figure 23.** predict_time comparison before and after RF optimization.

*4.4. FAMS Generalization Performance Test*

Among the metrics for evaluating the goodness of a FAMS, the generalization performance test of the FAMS is essential. In addition to the original Accuracy, Precision, Recall, F1_Score, Average, and predict_time, we have also added the AUC score indicator roc_auc_score to the FAMS generalization test performance evaluation metrics. On the other hand, we used a new dataset of over 330,000 items from CIC-DDoS2019, which we had not used before. The results of the two generalization performance tests are as follows: The generalization capability of the increased DDoS detection dataset is shown in the histogram Figure 24. Compared with the original 10M dataset, all performance metrics have been improved, including Accuracy by 0.00022, Precision by 0.00005, Recall by 0.00038, F1_Score by 0.00021, and Average by 0.00021, Table 12. The corresponding histogram is shown in Figure 24. As shown in Figures 25–27, the generalization performance test of the dataset with increased DDoS detection achieved excellent results.

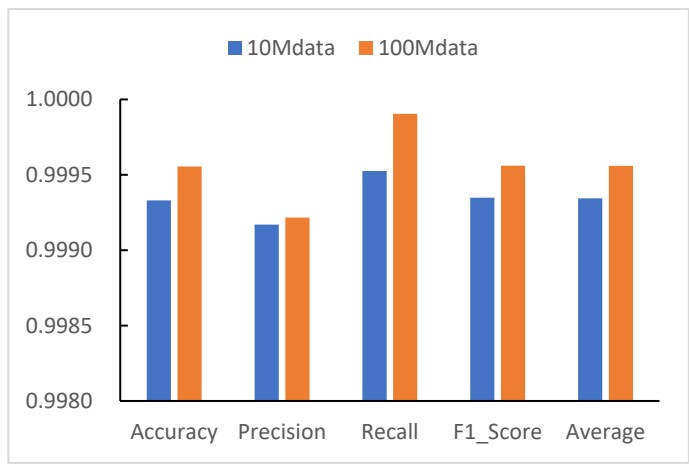

**Figure 24.** Performance comparison of increasing DDoS dataset.

**Table 12.** Performance metrics for generalization capability of increasing DDoS dataset.

|  | Accuracy | Precision | Recall | F1_Score | Average |
|---|---|---|---|---|---|
| 10Mdata | 0.9993 | 0.9992 | 0.9995 | 0.9993 | 0.9993 |
| 100Mdata | 0.9996 | 0.9992 | 0.9999 | 0.9996 | 0.9996 |
| Difference | 0.0002 | 0.0000 | 0.0004 | 0.0002 | 0.0002 |

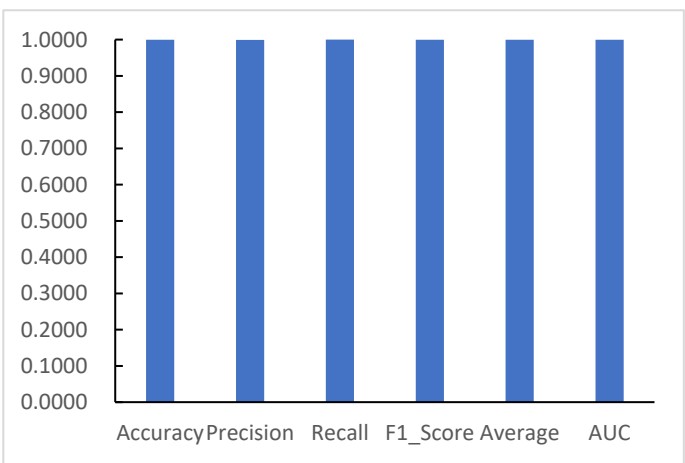

**Figure 25.** Performance demonstration of increasing DDoS dataset.

The FAMS performance was found to be equally strong after testing on a new 33M dataset from CIC-DDoS2019, as shown in Table 13 and Figure 28, compared to the original 10M. Compared to the original 10M dataset, the experiments on the 33M dataset show that our FAMS also improves in all performance metrics, with Accuracy improving by 0.00048, Precision improving by 0.00077, Recall improving by 0.00030, F1_Score improving by 0.00054, and Average improving by 0.00052. The histogram in Figure 29 of the dataset generalization capability test for changing DDoS detection corresponds to Figures 30 and 31 for the confusion matrix and ROC, respectively. Similarly, the generalization performance of the dataset with increased DDoS detection also yielded satisfactory results.

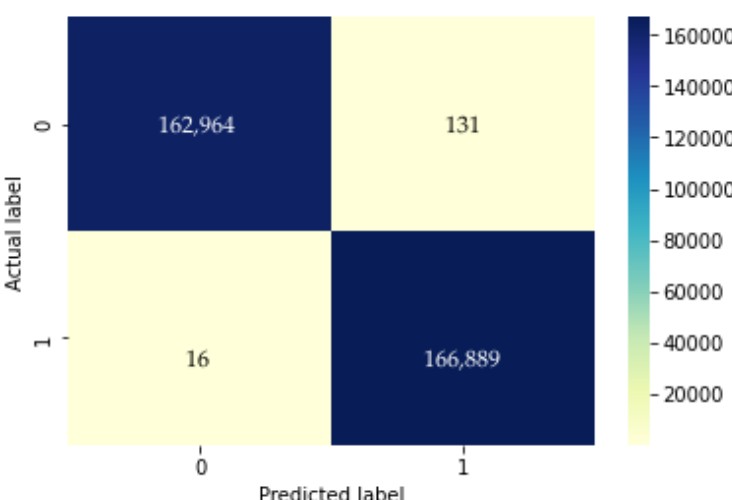

**Figure 26.** Increasing the DDoS dataset confusion matrix.

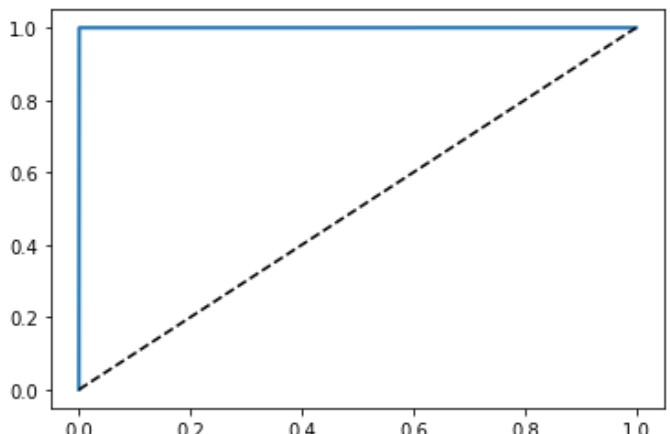

**Figure 27.** Increasing the DDoS dataset ROC.

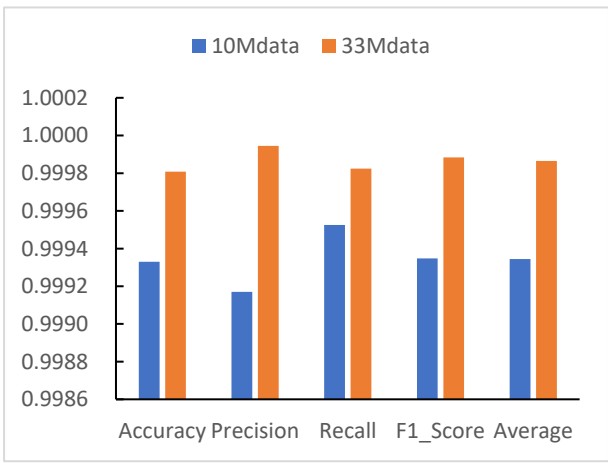

**Figure 28.** Performance comparison with changing DDoS dataset.

**Table 13.** Performance metrics for generalization capability of changing DDoS dataset.

|  | Accuracy | Precision | Recall | F1_Score | Average |
|---|---|---|---|---|---|
| 10Mdata | 0.9993 | 0.9992 | 0.9995 | 0.9993 | 0.9993 |
| 33Mdata | 0.9998 | 0.9999 | 0.9998 | 0.9999 | 0.9999 |
| Difference | 0.0005 | 0.0008 | 0.0003 | 0.0005 | 0.0005 |

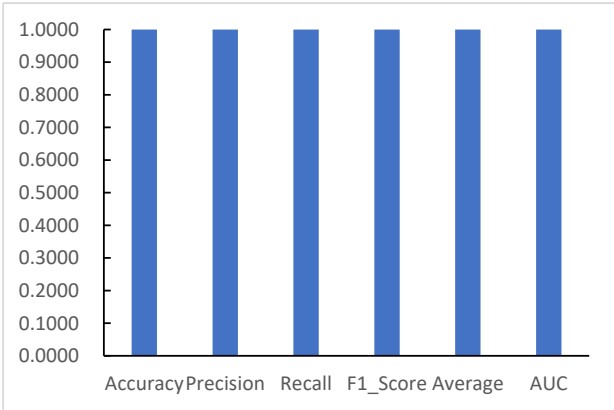

**Figure 29.** Performance demonstration with changing DDoS dataset.

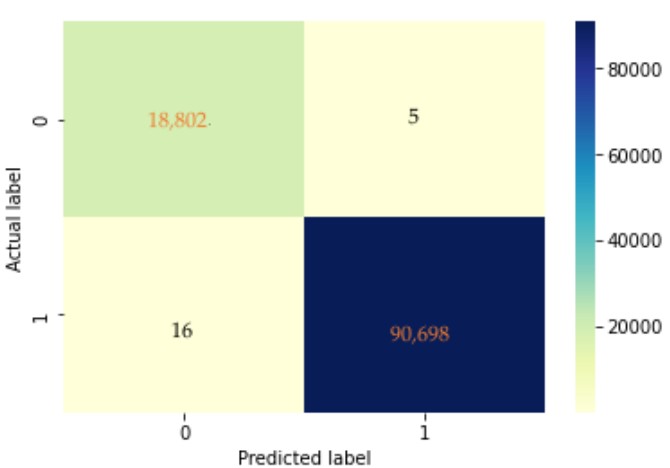

**Figure 30.** Changing the DDoS dataset confusion matrix.

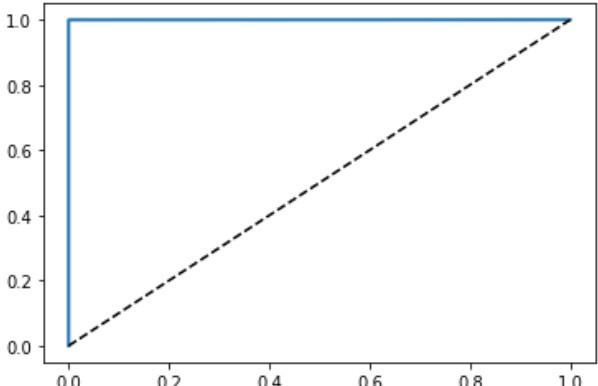

**Figure 31.** Changing the DDoS dataset ROC.

## 5. Conclusions

In this paper, we propose a framework based on feature and model selection, FAMS, and this framework contains a total of four phases. These are the data preparation phase, the feature selection (FS) phase, the model selection (MS) phase, and the RF optimization phase. Firstly, the data processing stage includes feature extraction, feature coding, missing value filling, outlier removal, and normalization operations. The data are pre-processed first. Then, in the feature selection phase, we propose a feature selection algorithm that includes filter, wrapper, and embedded in order to eliminate the bias and shortcomings of a single feature selection algorithm and generate 21 DDoS attack features. In the model selection phase, the RF was initially selected from GD, SVM, LR, RF, HVG, SVG, HVR, and SVR by the model selection algorithm. Finally, in the model optimization phase, the RF parameters max_samples, max_depth, and n_estimators were further optimized by the RF optimization algorithm. By testing the 100,000 CIC-IDS2018, CIC-IDS2017, and CIC-DoS2016 synthetic datasets, the results show that the detection Accuracy of this framework FAMS for DDoS attacks was 99.93%, Precision was 99.91%, Recall was 99.95%, and F1_Score was 99.93%, and the detection time of predict_time was only 0.1 s. All the results have achieved excellent performance in the same category. Moreover, the results show that the framework also shows excellent generalization performance by testing over 1 million synthetic datasets and over 330,000 CIC-DDoS2019 datasets. This framework, FAMS, achieves the goals of strong generalization capability, high prediction accuracy, and short prediction time for DDoS attack detection. For future work, our optimized RF model, with high accuracy, short prediction time, and high generalization performance for DDoS attack detection, is highly practical and can subsequently be deployed in production environments on distributed real-time detection systems in conjunction with big data technologies.

**Author Contributions:** Conceptualization, R.M. and X.C.; methodology, R.M.; software, R.M.; validation, R.M., X.C. and R.Z.; resources, R.Z.; writing—original draft preparation, R.M.; supervision, X.C. All authors have read and agreed to the published version of the manuscript.

**Funding:** This research was funded by National Natural Science Foundation of China, grant number U20A20179.

**Data Availability Statement:** https://www.unb.ca/cic/datasets/ids-2018.html (accessed on 13 August 2022). https://www.unb.ca/cic/datasets/ids-2017.html (accessed on 13 August 2022). https://www.unb.ca/cic/datasets/dos-dataset.html (accessed on 13 August 2022).

**Conflicts of Interest:** The authors declare no conflict of interest.

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
