# Peer review of "A DDoS Attack Detection Method Based on Natural Selection of Features and Models"

_electronics, doi:10.3390/electronics12041059_

Round 1

Reviewer 1 Report

The authors propose a framework based on feature and model selection for detecting Distributed Denial of Service attacks with the aim of identifying features and models with high generalisation capability, high prediction accuracy and short prediction time. The content of the paper is well written in abstract part; and the paper is very long.

I have some remarks:

1. Please first explain abbrivations then use it, for isntance FAM.

2. Remove one " model model," in abstract, and check the whole paper accordingly.

2. The formula in line 262 is not clear, please make it more clear. Also explain what is the role of this formula in your study there.

3. Line 289: Please explain what is argB.

4. Line 289: Please explain what are Xj and Bj.

5. Line 290: Please explain why do you need condition (3).

6. Lines 296, 298: Please expleain all items in the formulas.

7. Please be very careful about dots and all others, for instance line 316.

8. Lines 408, 412, 415: Please expleain all items in the formulas.

9. Write a longer version for the title of 3.4.4., 3.4.5

I am also not so sure that about the lenght of the paper.

Please explain in detail what is the aim of this length?

Reviewer 2 Report

The paper proposes a framework FAMS, a DDoS attack detection framework based on machine learning, which is divided into four phases, it was well introduced and designed with good discussion and demonstration of the results. Excellent tables and figures were displayed.

The title should replace the “ddos” with capital letters as "DDOS"

1. the Abstract should have no result value as shown in line (28 and 29).

2. Equation 7, should have clarification and description of the parameters as shown in line (392).

3. What is the confusion matrix for Figure 10, trying to add to the result and journal as it's not clearly discussed.

Round 2

Reviewer 1 Report

The current version is acceptable.